

**The formation processes and development characteristics of**
**sandbars due to outburst flood triggered by landslide dam**
**overtopping failure**
Xiangang Jiang[✉][1] · Haiguang Cheng[1] · Lei Gao[2] · Weiming Liu[3]
[1]College of Civil Engineering, Sichuan Agricultural University, Dujiangyan, Chengdu 611830,
China
[2]Key Laboratory of Ministry of Education for Geomechanics and Embankment Engineering,
Hohai University, Nanjing 210098, China
[3]Key Laboratory of Mountain Hazards and Earth Surface Process, Institute of Mountain Hazards
and Environment, Chinese Academy of Sciences, Chengdu 610041, China
Correspondence to: Xiangang Jiang (✉E-mail: jxgjim@163.com)
**Abstract**

13       Sandbars are an essential form of riverbed morphology which could be affected

by landslide dams. However, few studies have focused on the formation processes and
development characteristics of sandbars triggered by outburst flood. In such a way,
eight group dam failure experiments with 4 to 7 times of dam length movable bed is
carried out to study the temporal and spatial distributions of 25 sandbars along the
riverbeds, the sandbars geometric characteristics, and the influence of outburst flow
hydraulic characteristics on developments of sandbars. The results show that sandbars
are formed after peak discharge of outburst flow. The number of sandbars is 0.4 to 1.0
times the ratio of river bed length to dam length. Besides, sandbars have the
characteristic of lengthening towards upstream during the failure process. Sandbars'





upstream edges have a more extensive development than sandbars downstream edges.
The length of a sandbar along the channel changes faster than the sandbar's width and
height. The sandbars' length and width are about 10 to 80 and 1 to 7 times of average
height, respectively, and the average heights of sandbars are about 1 to 3.5 times the
maximum particle size. Sandbars' lengths make a more significant impact on sandbars'
volumes than widths and heights. It found that the Froude number has a significant
influence on the sediment carrying capacity. And the sediment concentrations in
volumes of the outburst flow at the upstream edges of all sandbars are greater than those
at the downstream edges of sandbars. Meanwhile, the sediment carrying capacities of
the outburst flow at the upstream edges of sandbars are smaller than those at the
sandbars' downstream edges. And the differences between the sediment concentrations
and the sediment carrying capacities determine the sedimentation or entrainment. The
results can reference the research on the river channel's geomorphological
characteristics affected by the outburst flood.
**Keywords**
Landslide dam · Overtopping failure · Sediment transport · Sandbar formation and
development
**1. Introduction**

Activities such as rainfalls and earthquakes often cause collapses, landslides,

which block the river to form a water retaining body similar to a reservoir dam, called
a landslide dam (Takahashi, 2007). According to statistics, 85 % of the dams were
destroyed within one year after formations, and more than 50 % of the dams were





damaged by overtopping (Costa and Schuster, 1988). Overtopping outburst floods are
extraordinarily destructive and seriously threaten people's personal and property safety.
Therefore, more and more scholars pay attention to the failure mechanisms and modes
of landslide dams and analyze outburst flood hydraulic characteristics and flood
evolution process (Pickert et al., 2011; Fan et al., 2012; Jiang et al., 2017, 2018, 2019a;
Zhang et al., 2019; Jiang and Wei, 2019b). Indeed, the outburst flood formed by
landslide dam failure carries loose materials in the channel during its evolution and
erodes and deposits along the channel. Sandbars are one typical landform formed during
the outburst flood evolution (Turzewski et al., 2019; Jiang and Wei, 2020; Wu et al.,
2020). Sandbars are shaped siltation bodies with exposed water surfaces formed by
rivers, lakes, and seashores (Chien et al., 1987). Moreover, sandbars are a feature of the
transition zone between aquatic and terrestrial, which have essential impacts on
transportation and species habitation using river corridors (Lin, 1990; Tracy-Smith et
al., 2012; Alexander et al., 2020). Consequently, sandbars have become the focus of
attention on river bedform and ecology.

At present, many researches about formations and developments of sandbars have

been conducted in natural rivers. Through field observations and indoor experiments,
sandbars' shapes and sizes can be observed intuitively, which is vital for understanding
formations and development characteristics of sandbars (Chien et al., 1987; Ashworth,
1996; Ashworth et al., 2000; Wright and Kaplinski, 2011; Demirci et al., 2014; Xie et
al., 2017; Alexander et al., 2020). For example, Chien et al. (1987) based on a large
number of field cases and data, and concluded that there are three basic types of





sandbars developments: (1) in the upstream backwater sections and the downstream
widening sections of the sandbars, sediments fall to promote the developments of
sandbars; (2) water flow erodes the front edges and sides of sandbars, and bends the
bars; (3) the protruding river core bedrock forces the flow to diverge and deposit the
sediments. Ashworth et al. (2000) through observing the nearly 1 km long sandbar of
the Jamuna River in Bangladesh, and sandbars' formation and development process
were analyzed. They pointed out that the cross-level formed by dunes and slip face
accretion at bar margins dominated developments of sandbars; Wright and Kaplinski
(2011) measured the three-dimensional flow structures and sandbars dynamics of the
two basins of the Colorado River in the Grand Canyon during the controlled flooding
of the Glen Canyon Dam. They found that the lateral reflux zone is conducive to fine
particle sediment deposition to form sandbars. Hooke and Yorke (2011) used remote
sensing images to analyze sandbars' dynamic evolution processes at multiple time
scales. They considered that the developments of sandbars are related to flow hydraulic
property. And they pointed out that analyzing the dynamic characteristics of sandbars
in rivers over a long period of time still needs more field data; Demirci et al. (2014)
obtained the dimensionless equation of sandbars' volumes through experimental data
using linear regression and nonlinear regression methods. The results showed that
experimental data are in good agreement with the proposed equation, but there was no
in-depth analysis of sandbars' other geometric features, and the relationship between
the geometric dimensions of sandbars was not clear; Xie et al. (2017) studied the
sandbars at the estuary of the Qiantang River and stated that the flow discharge played





a major role in sandbars' growths: when the flow discharge was large, the sandbars
would be eroded; when the flow discharge was small, the sandbars would be silted.

Some researchers have established mathematical models to simulate sandbars

growths and analyzed the development processes of sandbars. For example, Gao (1999)
believed that sandbars are the sedimentation results and used the hydrodynamic method
to derive the theoretical formula for sandbars' lengths. However, the method is not
suitable for unsteady flow, such as outburst flood caused by dammed lake overtopping
failure; Defina and Andrea (2003) established a two-dimensional finite element channel
morphological evolution model based on a non-cohesive river bed to simulate
formations and growths of sandbars. Using this model to study the impact of initial
disturbances on the initial flow field, which in turn affected sandbars growths; later,
Crosato and Mosselman (2009) simplified the physical mechanism of sandbars
formations and established a sandbar formation model. They considered that sandbars'
positions would change when the flow discharge changed or the riverside line was
eroded or deposited. And they proposed a quantitative method to predict the number of
sandbars in the river. But this model is suitable for rivers with a width height ratio less
than 100; Mueller and Grams (2018) coupled a simple morphological dynamics model
with flow and sediment concentration data, and it could reasonably predict sandbars'
volumes change. This method is aimed at the sandbars formed by the debris flow, but
the applicability of the sandbars formed by outburst flood remains to be investigated.

Sandbars formed after the landslide dam failure are caused by the strong unsteady

outburst flood. Kobayashi et al. (2010) established a two-dimensional morphological





dynamics model to study sandbars' growth processes under the action of unsteady flow.
And they discussed that flow unsteady property seemed to change the growth
mechanism of sandbars. Besides, for this type of sandbars, the upstream sediment is
mainly supplied by the dam material, which is different from other types of sandbars.
Until now, there is little field observation data of riverbed topography during landslide
dam breaching. As a result, questions remain regarding the formation processes and
development characteristics of the sandbars formed by outburst floods.
Overtopping failure is the most common failure mode of the landslide dam, so this
paper investigates the formation processes and growth characteristics of the sandbars
formed by the outburst flood due to landslide dam overtopping failure. This paper
focuses on the formation processes, the geometrical size characteristics of sandbars in
the downstream channel during the dammed lake's failure, and how the outburst flood
affects sandbars' developments. Firstly, through flume experiments, sandbars'
formation processes on the downstream channel under the dammed lake failure
condition were reproduced. Then, based on the experimental data, the growth
characteristics of sandbars' upstream and downstream edges were analyzed.
Furthermore, statistical analysis of sandbars geometrical dimensions at each moment
during the failure process, such as length, width, height, and volume, had been carried
out to obtain sandbars' size characteristics. Finally, by combining the hydraulic
characteristics of outburst flow at sandbars areas and sediment transport theory, the
sandbars' growth mechanisms were analyzed.



## 2. Experimental design

### 2.1 Model design and experimental materials

The longitudinal profiles of experimental landslide dams were trapezoidal and triangular. The trapezoidal dam height and crest width were both 0.3 m, and the triangular dam height was also 0.3 m. In the experiment, river bed slope angle $\theta$ was fixed at 10°, and the landslide dam upstream slope angle $\alpha$ was set to 40°, and the landslide dam downstream slope angles $\beta$ were set to five different values. The moveable bed was set downstream of the model dam, which had a length of 8 m. The downstream channel bed's length was about 4 to 7 times of dam length along the channel. The test parameters are shown in Table 1.

**Table 1** test parameters

| No. | Dam shape | $\beta$ (°) |
|-----|-----------|-------------|
| T1 | Trapezoid | 10 |
| T2 | Trapezoid | 15 |
| T3 | Trapezoid | 20 |
| T4 | Trapezoid | 25 |
| T5 | Trapezoid | 30 |
| T6 | Tringle | 10 |
| T7 | Tringle | 15 |
| T8 | Tringle | 20 |

Peng and Zhang (2012) proposed that landslide dam height ($H_d$), dam bottom width parallel to the channel ($W_d$), dam volume ($V_d$), and reservoir volume ($V_l$) are the key geometric parameters of landslide dam, and proposed a set of dimensionless numbers, $\frac{H_d}{W_d}$, $\frac{V_d^{1/3}}{H_d}$ and $\frac{V_l^{1/3}}{H_d}$, to verify whether the established dam model is consistent with the landslide dam in the field (Zhou et al., 2019). As the field data show that the $\frac{H_d}{W_d}$, $\frac{V_d^{1/3}}{H_d}$ and $\frac{V_l^{1/3}}{H_d}$ are ranged about 0.001 to 2, 0 to 40, and 0 to 20 for filed landslide dam (Zhou et al., 2019). Table 2 shows the dimensionless numbers of

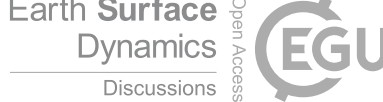

the experimental dams, which are all within the acceptable range of the field landslide
dams, indicating that the dams in the experiments are relatively close to field landslide
dams.
**Table 2** landslide dam parameters. The value of $\dfrac{H_d}{W_d}$ ranges from 0.1 to 0.3, and $\dfrac{V_d^{1/3}}{H_d}$ and $\dfrac{V_l^{1/3}}{H_d}$
both range from 1 to 2, which all fall within the acceptable range of values of the field landslide
dams (Zhou et al., 2019).

| No. | $H_d$(m) | $W_d$(m) | $\dfrac{H_d}{W_d}$ | $\dfrac{V_d^{1/3}}{H_d}$ | $\dfrac{V_l^{1/3}}{H_d}$ |
|---|---|---|---|---|---|
| T1 | 0.3 | 2.359 | 0.127 | 1.643 | 1.477 |
| T2 | 0.3 | 1.777 | 0.169 | 1.513 | 1.477 |
| T3 | 0.3 | 1.482 | 0.202 | 1.437 | 1.477 |
| T4 | 0.3 | 1.301 | 0.231 | 1.387 | 1.477 |
| T5 | 0.3 | 1.177 | 0.255 | 1.350 | 1.477 |
| T6 | 0.3 | 2.059 | 0.146 | 1.508 | 1.477 |
| T7 | 0.3 | 1.477 | 0.203 | 1.350 | 1.477 |
| T8 | 0.3 | 1.182 | 0.254 | 1.254 | 1.477 |

The dam materials used in this study were mixtures of sand and graves, with a
median particle size $D_{50}$ of 3.8 mm. Due to the flume space limitation, the maximum
sediment particle size was set to 20 mm. The riverbed was movable, which consisted
of the same material as the dam model. The thickness of the riverbed was set to 0.06 m.
The gradation curve of material particles' sizes is shown in Fig. 1.





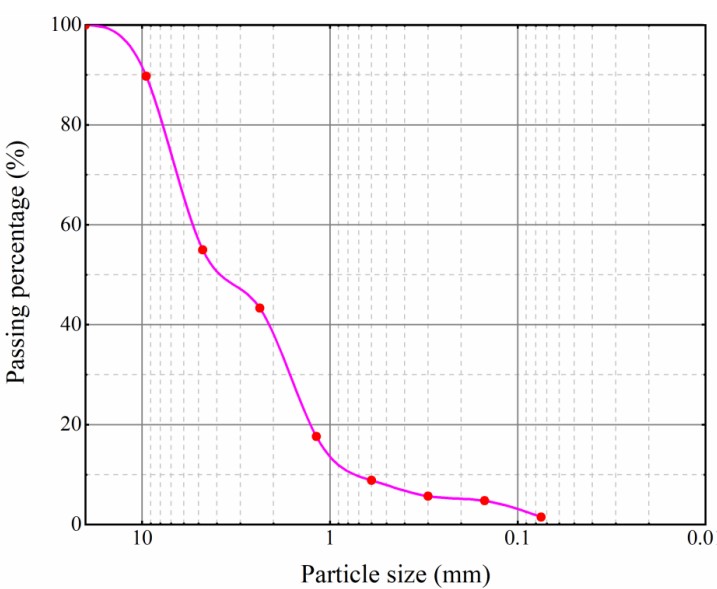


**Figure. 1** Gradation curve of the dam materials

**2.2 Experimental apparatus**
The experimental setups are shown in Fig. 2. The flume was 15 m long, 0.3 m
wide, and 0.6 m high. The flume slope was adjustable from 10 to 30°. One side of the
flume was transparent glass, and scale lines were drawn on the glass to facilitate
observation and recording of experimental phenomena. The inflow discharge was set
as 1.0 L s$^{-1}$. Under the control of the electromagnetic flowmeter, the error range could
be controlled within ±0.01 L s$^{-1}$. During the tests, the toe of the dam upstream slope
was set at 4.5 m away from the water supply tank. A baffle with a height of 6 cm was
set at the flume end as a boundary condition. Seven cameras were placed on the
transparent glass side of the flume, one camera was placed on the top of the dam, and
one camera was placed directly behind the flume. A total of nine cameras recorded the
whole experimental phenomena.



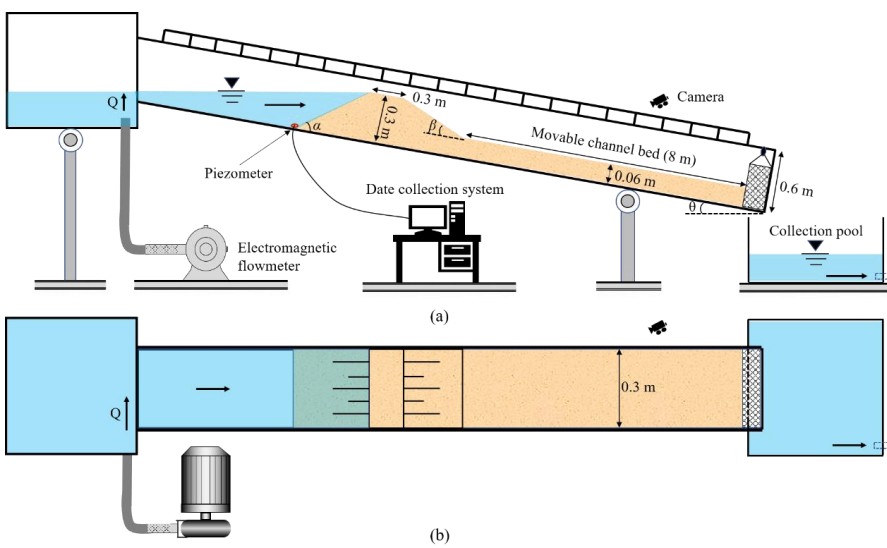

<p style="text-align:center">(a)</p>

<p style="text-align:center">(b)</p>


**Figure. 2** Experimental setups. (a) Front view of the flume. (b) Top view of the flume.

**2.3 Measurements**
In the experiment, the flow velocity was measured based on the reference object.
After the dam failure, a large number of small balls were continuously thrown into the
flume. Because the balls were of small mass and were eye-catching and easy to observe,
they would maintain the same movement state as the outburst flow under the flow's
drive. In a certain period, the balls' distance can be determined by the glass's scale lines
and then divide by the time to get balls' speeds, that is, the outburst flow velocities.
Flow depth could be read directly through the glass's scale lines, and the difference
between the flow surface elevation and the bed sand elevation represented flow depth.
Sandbars' lengths, widths, and heights could be obtained from the screen. It should
be noted that, due to the irregular shapes of the sandbars, the lengths of the sandbars
along the flume could be measured, but sandbars' widths and heights were different at
different locations. In this paper, the representative width and height values, which were





the arithmetic means along the channel, were used. Regarding the sandbars' volumes,
according to the actual sandbars' geometric characteristics, the sandbars were divided
into several parts, and then the volume calculation formula of the similar geometric
body was used to calculate the volume of each part respectively, and finally, the
sandbars' volumes were obtained by summing.
**3. Experimental results**
**3.1 Formation processes of sandbars**
The outburst flood due to the dam overtopping failure carried the downstream
channel's sediment and promoted formations and developments of sandbars. It showed
that three to four sandbars downstream the dam after the dam failure. Turzewski et al.
(2019) investigated the sandbars in the Yigong River triggered by the Yigong outburst
flood in 2000. They found that the number of sandbars is about 0.69 to 0.77 times the
ratio of river bed length to dam length for the sandbar frequent region. In this study,
sandbars were distributed in the 8 m length of the channel, which is 4 to 7 times of dam
length. It reflected the number of sandbars was 0.4 to 1.0 times the ratio of river bed
length to dam length. By comparing the experimental data and the field data of
Turzewski et al. (2019), it can be found that field data falls within the range of
experimental data. Experimental models took more influencing factors into account,
while the field data of Turzewski et al. (2019) only focused on the sandbars in the
Yigong River case, which is the reason for that field data falls within the range of
experimental data.



It took the T7 test as an example to analyze sandbars formation processes, as
shown in Fig. 3. Start timing when the flow just exceeded the dam crest, and at the
initial dam failure stage, the outburst flood carried the dam material to the dam
downstream slope (T=5 s). As the dam failed further, the flow discharge increased, and
outburst flood carried many dam materials to the channel bed (T=19 s). It should be
noted that although a large number of sediments were transported on the channel bed
before the peak discharge, no sandbar would be formed on the downstream channel bed.
After the moment of peak discharge, the flow discharge gradually weakened, and dam
materials were transported to the section near the dam toe. The flow could not transport
all the sediments away, and some sediments gradually silted down, then the first sandbar
occurred near the dam toe (T=30 s, the sandbar in the figure is marked with a blue
dotted line). After the first sandbar was formed, flow movement was changed. The
advancing flow bypassed the first sandbar, and the flow streamlines bent. Due to inertia,
the moving sediments no longer moved along the curved streamline but moved in the
original direction. On the opposite side of the first sandbar, sediments piled up to form
the second sandbar. With the first and second sandbars' existence, the flow streamline's
bending was more apparent, and flow moved along the "S" shaped path to the
downstream channel bed. It could be seen that there was a mutual feeding relationship
between sandbars and flow. That is, sandbars and flow influenced each other.
Similarly, the first and second sandbars affected the formation of the sandbar
downstream. Because of the accumulation and erosion of sediments, the channel bed's
sandbars kept growing, and sandbars' locations and geometric dimensions were



changed. For example, when T=33 s, more and more sediments were deposited on the
upstream sandbars' edges, the sandbar near the dam toe continued to grow, and the
upstream sandbar's volume increased. When the dam was failed entirely, the sandbars
had changed significantly compared to the initial sandbars, making the channel bed
topography changed significantly. Eventually, sandbars were scattered on both sides of
the flume, forming a meandering channel downstream (T=40 and 47 s). This
phenomenon is in good agreement with the field sandbars along the Yigong river (Wu
et al., 2020).
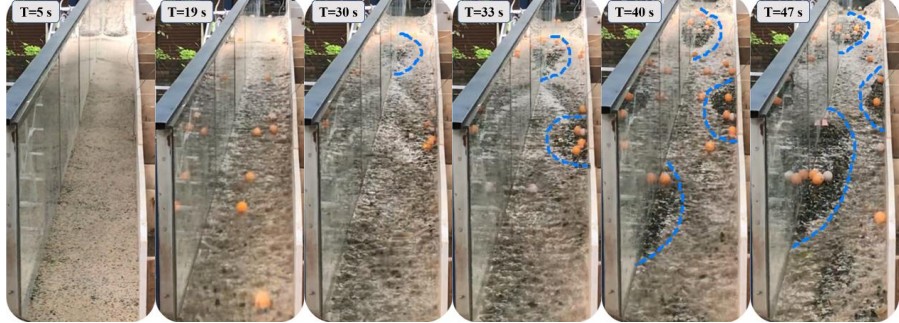
**Figure. 3** The riverbed morphology at six different moments during the sandbars' formations and

growths for the T7 experiment. The sandbars in the figure are marked with blue dotted lines.

**3.2 Position characteristics of the sandbars' edges**

Figure. 4 shows sandbars' locations on the channel bed during the dam failure. The

red lines in the figure represent the sandbars, and the orange rectangles represent the
flumes. Figure. 4 can clearly show the formation sequences of sandbars at different
locations. That is, sandbars were formed first near the dams, and the farther from the
dam toe, the later the sandbar was formed, which is consistent with the content of Sect.
3.1. Sandbars near the downstream dam toes are all located on the dam breach side



across the river. This characteristic has also been found in Chen et al. (2015).
According to the sandbars' formation sequences, the channel bed's sandbars were
divided into three types: the sandbar near the dam toe, the sandbar near the middle
reaches, and the sandbar near the bed end. And the characteristics of the position
changes of the sandbars' upstream and downstream edges were analyzed, respectively.
Figure 4 that the upstream edges of the sandbars near the dam toes in the eight group
experiments basically moved upstream with time. But the movement directions of the
downstream edges of the sandbars near the dam toes showed diversity: in the two tests
of T2 and T5, the sandbars' downstream edges moved toward the dam toes, from a
distance from the downstream toe of 3.3 to 2.9 m and 3.7 to 3.5 m respectively, as
shown in Fig.4(b) and (e); in the tests of T1, T6, T7, and T8, the sandbars' downstream
edges first moved away from the dam toes and then moved toward the dam toes, and
the downstream edges move forward compared to the original location. However, the
distance they moved is 0.15 to 0.7 m, which is small as shown in Fig.4(a), (f), (g), and
(h); in the experiments of T3 and T4, the sandbars' downstream edges positions
remained almost unchanged, see Fig.4(c) and (d). However, no matter how the
downstream edge positions of the sandbars near the dam toes changed, the results of
the eight tests have a common feature: compared with the position when the sandbars
were formed, the downstream edges moved less distance, and the amount of movement
was much smaller than those of sandbars' upstream edges. The lengths of the sandbars
near the dam toes increased with the failure time. It can be seen that the sediments on
the sandbars' upstream edges played a greater role in the length developments of the





sandbars near the dam toes.

(a)

(b)

(c)

(d)





(e)
(f)
(g)
(h)



**Figure. 4** The sandbars' locations during the dam failure: (a) sandbars' locations of the T1 test; (b)
sandbars' locations of the T2 test; (c) sandbars' locations of the T3 test; (d) sandbars' locations of
the T4 test; (e) sandbars' locations of the T5 test; (f) sandbars' locations of the T6 test; (g) sandbars'
locations of the T7 test; (h) sandbars' locations of the T8 test. The red lines in the figure represent
the sandbars, and the orange rectangles represent the flumes. The numbers at both ends of the red
lines represent the distances between the two edges of sandbars and the dam toe, that is, the distances
between the upstream and downstream edges of sandbars and the dam toe.
Growth characteristics of the upstream and downstream edges of the sandbars near
the bed ends were similar to those of the sandbars near dam toes. That is, upstream
edges grew toward dam toes, and the upstream edges move more extensively than the
downstream edges. And sandbars downstream edges almost remained at the initial
location. Sandbars' lengths gradually increased throughout the process of dam failure.
Compared with the sandbars near the dam toes, the sandbars' movements in other reach
were smaller. The distance between the sandbars in middle and end reach is smaller
than the distance between sandbars near dam toe and adjacent sandbars.
The dam downstream slope and longitudinal section shape also influenced the
sandbars. The largest movement distance for upstream edges of sandbars near the dam
toe moved was 1.8 m, and for downstream edges of sandbars was 0.7 m with a
downstream slope angle of 10° for trapezoidal shape models. It was the smallest
movement distance for the upstream edge and the largest movement distance for the
downstream edge for the trapezoidal shape models. The sandbar's final length was 1.2
m, which was the shortest among the downstream slope angle from 10 to 30°. However,





sandbar length varied small with a maximum difference of 0.4 m when the angle
increased from 15 to 30°. The lengths of sandbars in the middle and end reach were
also the smallest for the 10° downstream slope. However, the distance between the
sandbars was largest for the 10° downstream slope, and this is due to the smaller
outburst flood discharge and capacity of bedload for the 10° downstream slope. The
overlapping phenomena existed along the channel for sandbars with large downstream
slope, such as the sandbars in the middle and end reach at 60 s for T2. For the triangular
shape of the dam, the dam volume was the main factor influencing sandbars'
developments. It could be demonstrated from two sides: one is the number of sandbars
in the test, which dam downstream slope is 10° and with a larger dam volume, was
more than the number of sandbars for 15 and 20° downstream slope dam models; the
other is the lengths of sandbars in middle and end reaches become smaller with the
increasing downstream slope from 10 to 20° (i.e., with decreasing dam volume).
**3.3 Characteristics of the sandbars' geometric sizes**
Corresponding to Sect. 3.2, Fig. 5 shows that the lengths of the sandbars near the
dam toes were longer than other sandbars' lengths, and the sandbars near the dam toes
appeared first. Because the sandbars near the dam toes were closer to the dams, when
the flow carried a large number of sediments from the dam downstream slopes to the
channel beds, the slopes decreased, and a large number of sediments accumulated
around the sandbars near the dam toes to promote sandbars' developments. Sufficient
incoming sand from the upper reach made the lengths of the sandbars near the dam toes



larger than the other sandbars' lengths. For all the sandbars, their lengths were largest
in the whole process, followed by widths, and finally were heights. Sandbars' lengths
had a growing trend, and their growth rates were more significant than growth rates of
widths and heights. The sandbars' shapes were irregular during the entire dam failure
process, which is similar to the field sandbars (Wu et al., 2020). The average values of
the widths and heights of the entire sandbars were selected as the parameters reflecting
the characteristics of sandbars' widths and heights shown in Fig. 5. From the figure, we
can know that sandbars' widths changed more drastically than the sandbars' heights,
which is mainly because sandbars' heights were significantly affected by outburst flow
depth. In most cases, flow depth was less than the heights of sandbars, the sediments
mostly accumulated at the sandbars' edges and waists, and could not "climb up"
sandbars' tops; in addition, the reduction of flow depth was not large enough, so the
sandbars' heights did not change much. The variations of widths and heights both
increase slowly with time and then tended to be stable values.



Earth **Surface**
**Dynamics**
Discussions

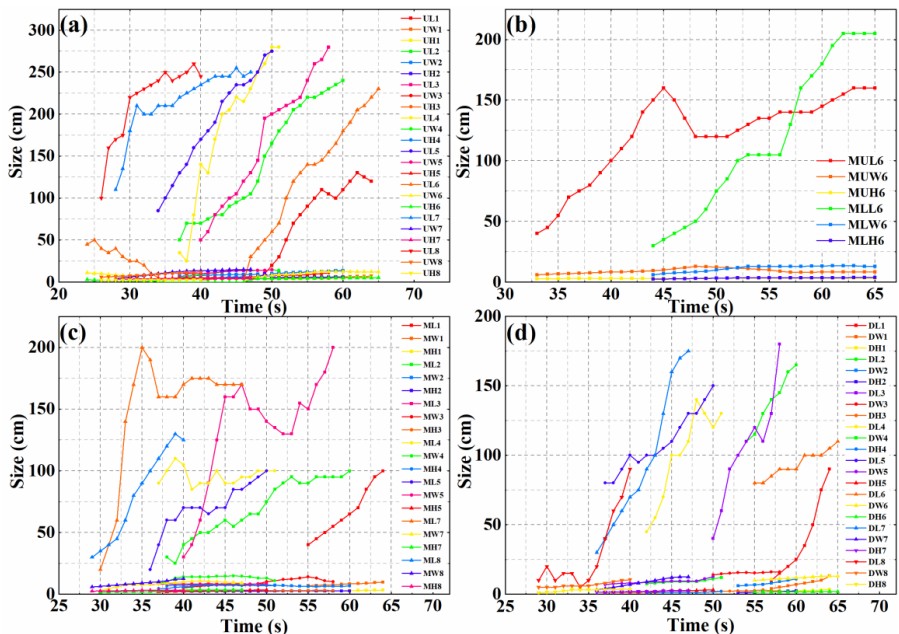

**Figure. 5** The lengths, widths, and heights of the sandbars: (a) sizes of the sandbars near the dam

toes; (b) sizes of the sandbars near the middle-upper and middle-lower reaches; (c) sizes of the

sandbars near the middle reaches; (d) sizes of the sandbars near the bed ends. Notation: ULi, UWi,

and UHi represent the length, width, and height of the sandbar near the dam toe of the Ti test,

respectively. For example, UL1 indicates the length of the sandbar near the dam toe of the T1 test;

MULi, MUWi, and MUHi represent the length, width, and height of the sandbar near the middle

and upper reaches of the Ti test, respectively. For example, MUL1 indicates the length of the sandbar

near the middle and upper reaches of the T1 test; MLi, MWi, and MHi represent the length, width,

and height of the middle sandbar of the Ti test, respectively; MLLi, MLWi, and MLHi represent the

length, width, and height of the sandbar near the middle and lower reaches of the Ti test, respectively;

DLi, DWi, and DHi represent the length, width, and height of the sandbar near the bed end of the Ti

test, respectively.

When the amounts of sediments deposited on sandbars were larger than the



quantities of eroded sediments, sandbars' volumes became larger. Otherwise, sandbars'
volumes would decrease or remain at a stable level. Figure. 6 reveals sandbars' volume
characteristics during the dam failure. Most of the 25 sandbars gradually increased in
volume, indicating that the amounts of outburst flow erosions in the sandbars' vicinities
were less than the amounts of siltation during the entire outburst process. The volumes
were about 0.018 to 0.142, 0.009 to 0.055, and 0.014 to 0.055 times of the initial dam
volumes for the sandbars near dam toes, the sandbars near the middle reaches, and the
sandbars near the end reach, respectively. It indicates that sandbars' total volumes in the
downstream channel of 4 to 7 times dam length to the initial dam volumes are about
0.009 to 0.142. By referring to Figs. 5 and 6, the sandbars' volume characteristics were
consistent with the sandbars' length characteristics. And because the widths and heights
developed in small change, sandbars' volumes were mainly controlled by sandbars'
lengths.

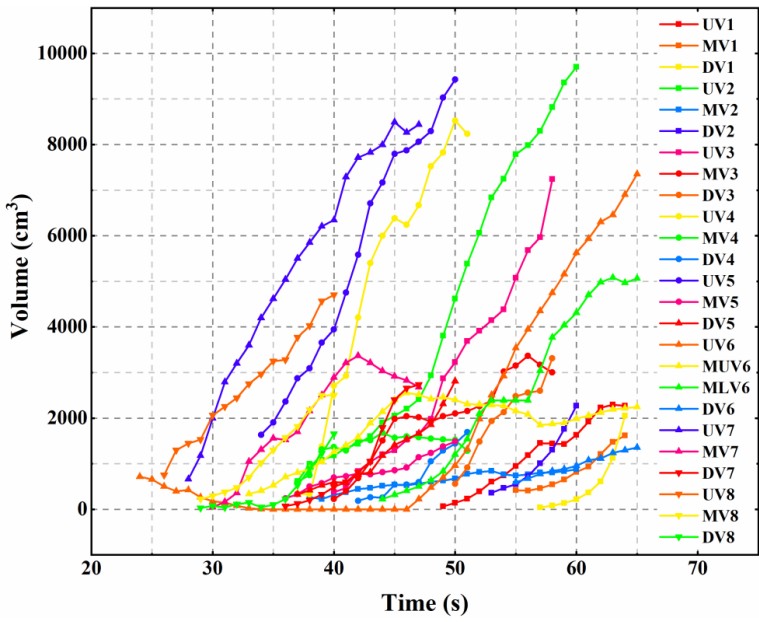


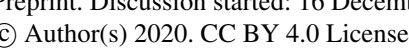

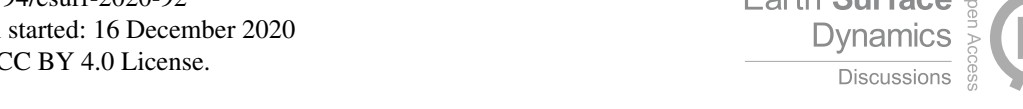

**Figure. 6** Volumes of sandbars. Notation: UVi, MVi, DVi, MUVi, MLVi represent the sandbar's
volume near the dam toe, the sandbar near the middle reaches, the sandbar near the bed end, the
sandbar near the middle-upper reaches, and the sandbar near the middle-lower reaches, respectively.
For example, UV1 means that the sandbar's volume near the dam toe of the T1 test.

Jiang and Wei (2020) discussed the relationships between the lengths and the

maximum widths and heights of sandbars when the dam was failed entirely, but the
relationships with sandbars lengths, average widths, and heights had not been involved.
It found that the average heights after the dam failure were about 1 to 3.5 times the
maximum grain size. The ratios of lengths to average heights were basically between
10 to 80, and the rate of average widths to average heights were basically between 1 to
7 (Fig. 7).

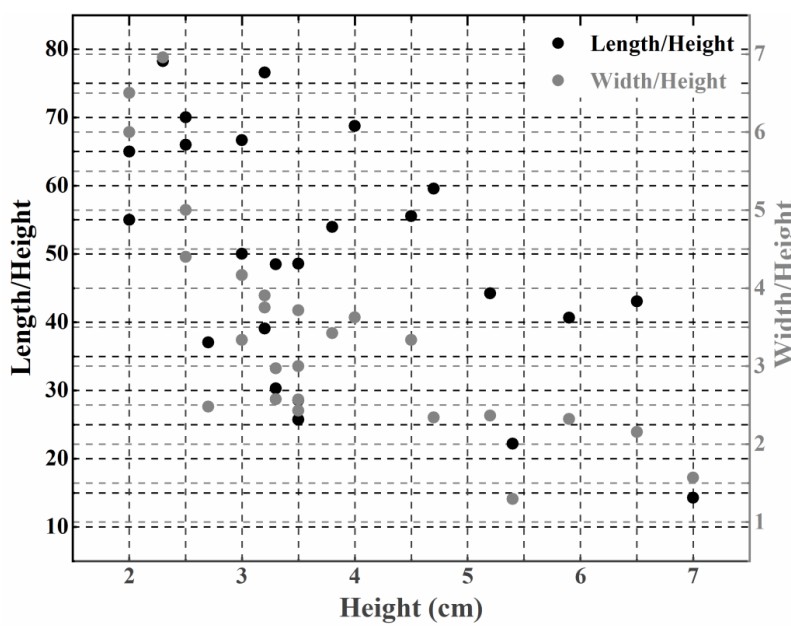


**Figure. 7** The ratios of length and height, width and height of the sandbar at the end of the dam
failure



## 4. Hydraulic characteristics of flow at the edges of the sandbars


Outburst flow influences the sandbars' formations and growths directly, and the
existence of sandbars will also affect the outburst flow hydraulic characteristics. In
order to understand how outburst flow affects sandbars' formations and developments,
it is necessary to explore the outburst flow hydraulic characteristics.
As stated in Sect. 3, the sandbars' lengths are the most critical parameters affecting
the sandbars' volumes, and sandbars' lengths are the most sensitive to the flow.
Therefore, analyzing hydraulic parameters at the sandbars' upstream and downstream
edges is helpful to understand the impact of the outburst flow on sandbars growths.
Sediment concentration in volume is an important physical parameter of sediment-
laden flow and is closely related to sandbars growths. The concentration calculation
method of Laursen (1958) was used to analyze the sediment concentrations in volumes
at the sandbars' upstream and downstream edges. In order to facilitate the comparison
of the sediment concentrations in volumes at the sandbars' edges, the average values of
the sediment concentrations for the 25 sandbars' edges were taken (from the moment
the sandbars were formed to the moment the dam was failed entirely), as shown in Fig.

8.

From Fig. 8, it reflects that average concentrations of the upstream edges of the
sandbars near the dam toes are the largest, mainly because this location was close to the
dam. Flow transported the dam materials to the vicinities of the sandbars near the dam
toes, and the amounts of sediments transported were more than other parts of the
channel bed. The sediment concentration of flow along the channel bed gradually



decreased. The part of the sediments that caused the sediment concentration decreased
to participate in sandbars' formations and growths. From the perspective of the entire
sediment concentration variation range, there was a little difference between the
concentrations at the upstream edges of the sandbars near the dam toes and the
concentrations at the downstream edges of the sandbars near the bed ends, indicating
that only a small part of sediments participated in the developments of the sandbars.
The sediment concentrations of flow at the upstream and downstream edges of all
sandbars had the same characteristic. The concentrations of flow at the upstream edges
of sandbars were larger than that at the downstream edges. This was mainly because
when the flow goes through the sandbars areas, some sediments deposit on the sandbars'
upstream edges and abdomens, causing sandbars growths.

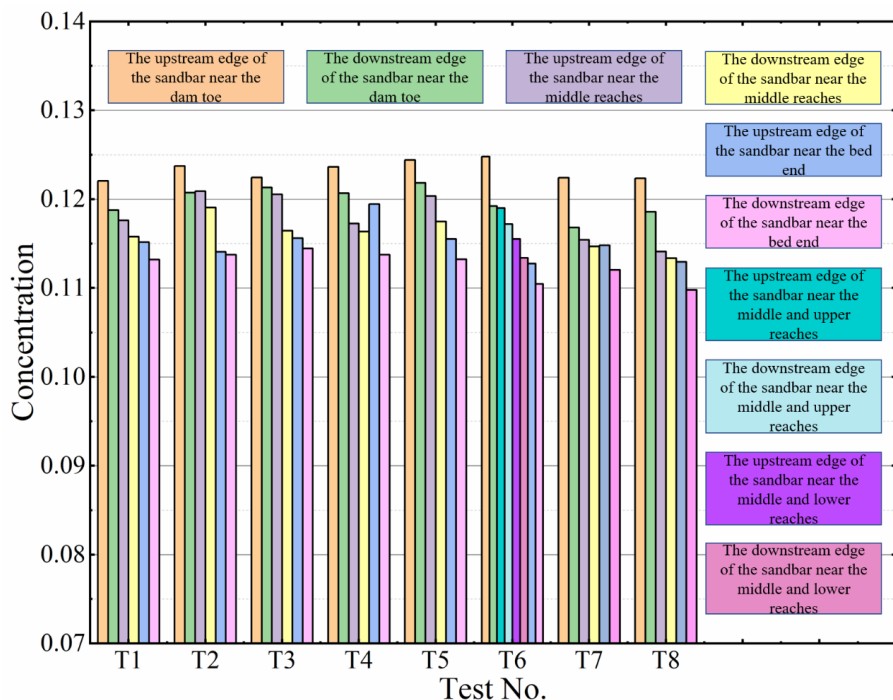

**Figure. 8** Sediment concentrations in volumes at edges of sandbars




Earth **Surface**
Dynamics
Discussions
EGU

The ratio of the inertial force to the gravity of the outburst flow can be reflected
by the Froude number ($Fr$). The equation for calculating the Froude number is as follow:

$$Fr = \frac{u}{\sqrt{gh}},$$ (1)

where $u$ is the flow velocity, m s$^{-1}$; $g$ is the acceleration of gravity, m$^2$ s$^{-1}$; $h$ is the flow
depth, m.
In order to facilitate the comparison of the flow Froude numbers at different
locations, the flow Froude numbers at the upstream and downstream edges of the 25
sandbars were taken as the average values over time of the entire dam failure process
(from the moment the sandbar was formed to the moment when the dam was failed
entirely), as shown in Fig. 9. It can be found that although the velocity and depth of
flow in the late period of the peak discharge gradually decreased, the average Froude
numbers of flow at the upstream and downstream edges of the sandbars are greater than
1, and it reflects the inertia effect is strong in these locations. If the streamline is
changed in these locations, the particles in the water may move laterally or the original
path causing sedimentation or erosion.

Earth **Surface**
**Dynamics**
Discussions

EGU

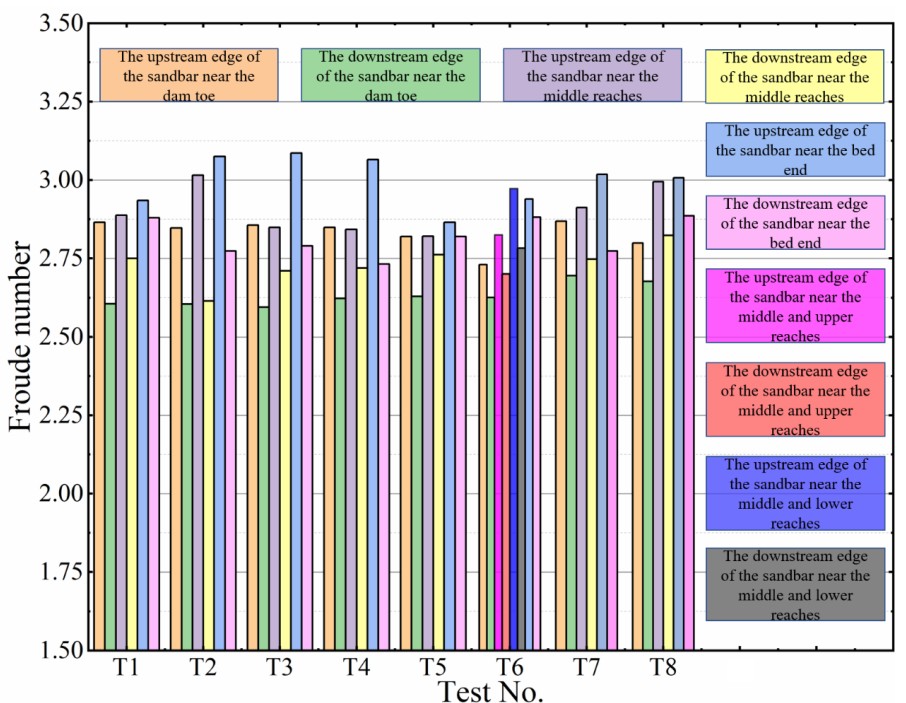

**Figure. 9** Froude numbers of outburst flow at the edges of sandbars
The flow sediment carrying capacity indicates the amounts of sediments that can
be carried through a river section under certain flow and boundary conditions. For these
experiments, sandbars' formations and growths mainly depended on the accumulation
of bedload, so we focused on the sediment carrying capacity of bedload. The calculation
equation of bedload sediment carrying capacity is

$$c_e = \frac{q_b}{hu}, \qquad (2)$$

where $c_e$ is bedload sediment carrying capacity, $q_b$ is the unit-width bedload transport
rate, and it can be calculated using the MPM equation (Meyer-Peter, 1948)

$$q_b = 8\sqrt{(s-1)gd^3}\,(\theta - \theta_c)^{1.5}, \qquad (3)$$

where $\theta$ is the Shields number, which can be obtained according to Eq. (4), and $\theta_c$ is the
critical Shields number. Referring to Misri et al. (1984), $\theta_c$ is taken as 0.03 in this paper;

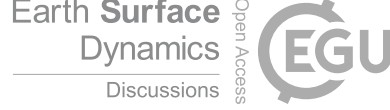


$s$ is the submerged specific gravity of sediment, which can be calculated according to
the Eq. (5); $d$ is the particle size of the sediment, m.

$$\theta = \frac{u_*}{(s-1)gd},\qquad(4)$$

$$s = \frac{\rho_s}{\rho_w},\qquad(5)$$

where $u_*$ is the frictional flow velocity, m s$^{-1}$; $\rho_s$ is the weight of sediment, and $\rho_w$ is the
weight of water.
The volume of sediments that can be carried by the flow in the flow sections could
be obtained with the above equations. Similarly, taking the average values of the
sediment carrying capacities of flow (from the moment of sandbars formations to the
moment of complete failure of the dam) for analysis, as shown in Fig. 10.

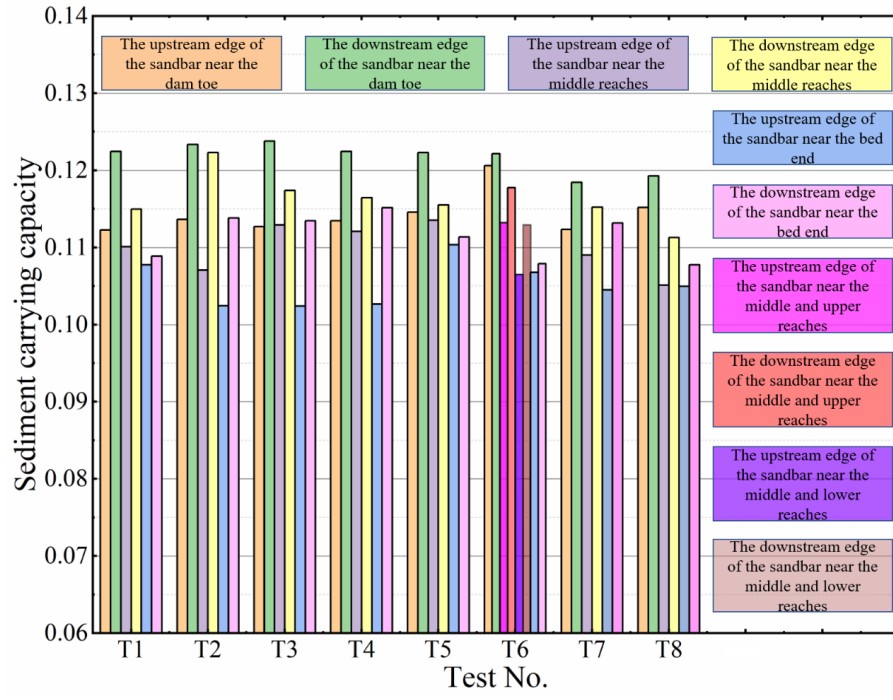


**Figure. 10** Sediment carrying capacities of outburst flow at the edges of sandbars





It shows that, from the whole process's perspective, comparing the sediment
carrying capacities at the upstream and downstream edges of different sandbars, the
sediment carrying capacities decrease along the downstream channel bed. Taking the
T1 test for example, the sediment carrying capacity of the flow at the upstream edge of
the sandbar near the dam toe was larger than the sediment carrying capacity at the
upstream edge of the sandbar near the middle reaches. The sediment carrying capacity
at the downstream edge of the sandbar near the dam toe was larger than the sediment
carrying capacity at the downstream edge of the sandbar near the middle reaches. The
characteristic indicates that the outburst flow erosion effect gradually weakened with
the distance from the dam. Compared to the sediment carrying capacities at the
upstream and downstream edges of the same sandbar, it can be found that the sediment
carrying capacity at the upstream edge was smaller than that at the downstream edge,
but the difference was not large. Through combining Figs. 9 and 10, it can be found that
there is a relationship between the sediment carrying capacity and the Froude number.
That is, when the Froude number increases, the sediment carrying capacity will
decrease; when the Froude number of the flow decreases, the sediment carrying
capacity will increase.
**5. The influence of outflow transportation capacity on development of**
**sandbars' lengths**
Sandbar length is the predominant factor to control the volume. The transportation
condition of outflow at the sandbars' edges determines sandbar growth: for the sandbars'



upstream edges, if the sediment concentrations in volumes are greater than the sediment
carrying capacities of the flow, sediments are accumulated, and the upstream edges will
extend toward the dam toes. And suppose the sediment concentrations in volumes are
less than the flow sediment carrying capacities. In that case, the upstream edges of
sandbars will be in a state of erosion, and the sandbars' upstream edges will extend far
away from the dam toes. As for the sandbars downstream edges, when the sediment
carrying capacities of the flow are smaller than the sediment concentrations, it means
that the flow cannot take away all sediments. Sediments will deposit, and the
downstream edges of sandbars will extend far from the dam toes; when the sediment
carrying capacities are larger than the sediment concentrations, the downstream edges
of the sandbars will be in an eroded state, and the sediments are carried by flow to the
downstream channel bed, and the sandbars' downstream edges will extend toward the
dam toes; when the sediment carrying capacities of the flow are equal to the sediment
concentrations, then the flow at sandbars' downstream edges will be in an equilibrium
sediment transport state. Figure. 11 shows the relationships between the sediment
concentrations in volumes and the sediment carrying capacities at the sandbars
upstream and downstream edges. It can be seen that the differences between the
sediment concentrations in volumes and the sediment carrying capacities ($c$-$c_e$)
fluctuate during the whole process of sandbars developments. Through referring to Figs.
4 and 11, the two pictures are highly consistent. When the ($c$-$c_e$) in Fig. 11 is greater
than 0, the sandbar's upstream edge point migrates upstream, or the downstream edge
point of the sandbar migrates downstream in Fig. 4. When the ($c$-$c_e$) is less than 0, the



Earth **Surface**
**Dynamics**
Discussions

sandbar's upstream edge point migrates downstream, or the downstream edge point of
the sandbar migrates upstream.

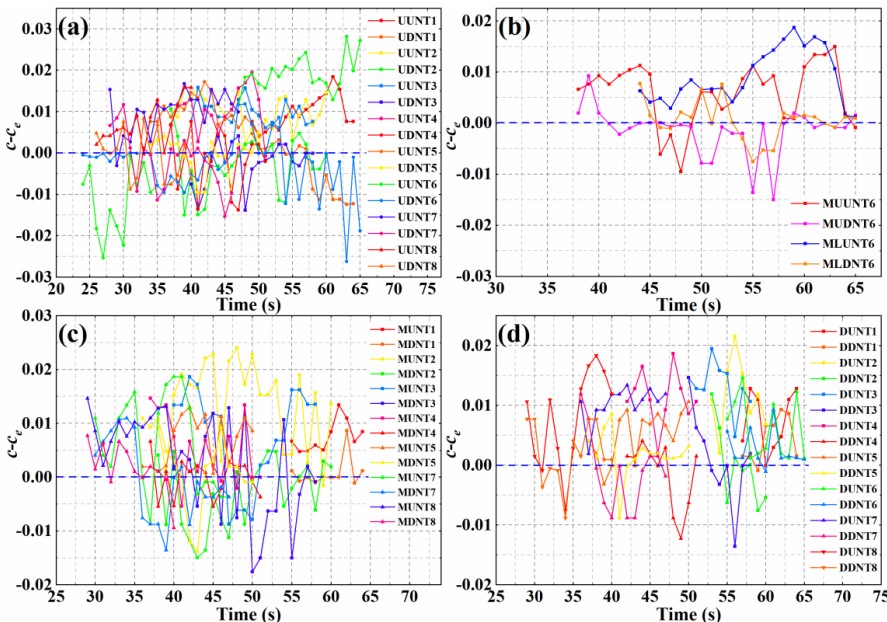


**Figure. 11** The difference of the sediment concentrations in volumes and the sediment carrying
capacities ($c$-$c_e$) at upstream and downstream edges of sandbars: (a) the ($c$-$c_e$) at the edges of the
sandbars near the dam toes; (b) the ($c$-$c_e$) at the edges of the sandbars near the middle and upper
reaches; (c) the ($c$-$c_e$) at the edges of the sandbars near the middle reaches; (d) the ($c$-$c_e$) at the edges
of the sandbars near the bed ends. Notation: $c$ is the sediment concentration in volume; $c_e$ is the
sediment carrying capacity of flow; UUNTi, UDNTi represent ($c$-$c_e$) at the upstream and
downstream edges of the sandbar near the dam toe of the Ti test. For example, UUNT1 represent
the value of ($c$-$c_e$) at the upstream edge of the sandbar near the dam toe of the T1 test; MUUNTi,
MLDNTSi represent ($c$-$c_e$) at the upstream and downstream edges of the sandbar near the middle
and upper reaches of the Ti test; MLUNTi, MLDNTSi represent ($c$-$c_e$) at the upstream and
downstream edges of the sandbar near the middle and lower reaches of the Ti test; MUNTi, MDNTSi





represent ($c$-$c_e$) at the upstream and downstream edges of the sandbar near the middle reaches of the
Ti test; DUNTi, DDNTi represent ($c$-$c_e$) at the upstream and downstream edges of the sandbar near
the bed end of the Ti test. (i=1 to 8)

The sums of ($c$-$c_e$) at the sandbars' upstream and downstream edges are used as

the criterion for judging the sandbars' length variation. The relationships between the
sums of ($c$-$c_e$) and zero determine the increase or decrease of sandbars' lengths. Suppose
the sums of (c-ce) are greater than zero. In that case, it means that the outburst flow
cannot transport all the sediments. The excess sediments are deposited in the sandbars
areas, corresponding to the increase in sandbars' lengths and volumes; otherwise,
sandbars' lengths and volumes are reduced. Figure. 12 shows the relationships between
the sums of ($c$-$c_e$) at the sandbars' edges and 0. By combining Figs. 12 and 6, it can be
seen that when the sums of ($c$-$c_e$) are greater than 0, sandbars' lengths and volumes are
increased. It reveals that the relationship between the sums of ($c$-$c_e$) and 0 can be used
to judge the trend of sandbars' lengths and volumes.



Earth **Surface**
**Dynamics**
Discussions

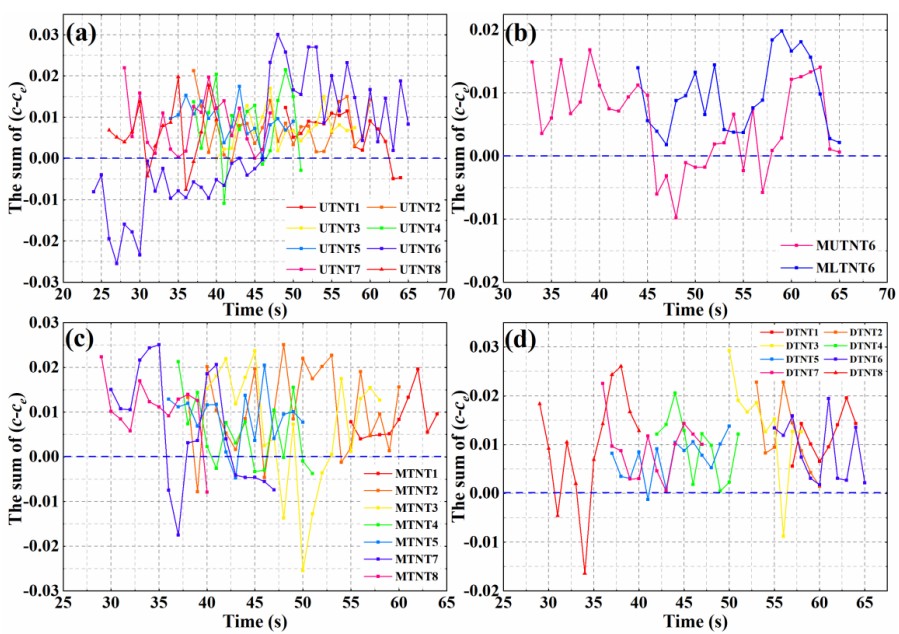

**Figure. 12** The sums of ($c$-$c_e$) at the upstream and downstream edges of sandbars: (a) the sums of

($c$-$c_e$) at the upstream and downstream edges of the sandbars near the dam toes; (b) the sums of ($c$-

$c_e$) at the upstream and downstream edges of the sandbars near the middle-upper reaches, and the

sandbars near the middle-lower reaches; (c) the sums of ($c$-$c_e$) at the upstream and downstream

edges of the sandbars near the middle reaches; (d) the sums of ($c$-$c_e$) at the upstream and downstream

edges of the sandbars near the bed ends. Notation: for the Ti test, UTNTi, MUTNTi, MLTNTi,

MNTTi, DTNTi respectively represent the sum of the ($c$-$c_e$) at the upstream and downstream edges

of the sandbar near the dam toe, the sandbar near the middle-upper reaches, the sandbar near the

middle-lower reaches, and the sandbar near the middle reaches, and the sandbar near the bed end.

(i=1 to 8)

## 6. Conclusion

In this paper, a downstream moveable bed with 4 to 7 times the length of dam



length along the channel was set, and through eight flume experiments, 25 sandbars
were formed downstream channel caused by overtopping flow. The sandbars
development characteristics and the influences of hydraulic parameters on sandbars
were also analyzed. The main conclusions are as follows.
(1) The number of sandbars is 0.4 to 1.0 times the ratio of river bed length to dam
length. Sandbars first appeared near dam toes located on the dam breach sides across
the rivers. Inertia force made sediment accumulate on the opposite banks of the channel
bed, resulting in the formations of sandbars downstream. Meanwhile, it has the
characteristic that the farther away from the dam, the later the sandbar formation.
During the evolution of outburst flow, the sandbars' upstream edges are mainly in
siltation states. The sandbars' lengths increase with failure time, mainly caused by
sandbars' upstream edges move upstream. The downstream edges develop slowly and
basically near the initial positions. And the developments of sandbars downstream
edges are much smaller than the developments of sandbars' upstream edges.
(2) During dam failure, the lengths varied faster than the widths and heights of
sandbars. And the lengths along the river are the largest, followed by widths, and finally
are sandbars' heights after the dam failure. The average sandbars' heights are about 1 to
3.5 times the maximum particle size. The sandbars' lengths are about 10 to 80 times the
average heights, and the average widths are 1 to 7 times the average heights. The lengths
mainly control the sandbars' volumes. The ratio of sandbars' total volumes in the
downstream channel of 4 to 7 times dam length to initial dams' volumes are about 0.009
to 0.142.




(3) The impact of outburst flow on sandbars is mainly manifested by the sediment
concentration and the sediment carrying capacity. During the entire dam failure process,
the sediment concentrations at sandbars' upstream edges are greater than that on the
downstream edges. The Froude number has a significant influence on the sediment
carrying capacity. When the Froude number increases or decreases, the sediment
carrying capacity decreases or increases accordingly. The sediment carrying capacities
at the sandbars' upstream edges are smaller than those at the sandbars' downstream
edges. The characteristics of sediment concentrations and sediment carrying capacities
at sandbars' edges cause sandbars to develop upstream.
(4) The formation processes and development characteristics of sandbars from the
perspective of flow transporting sediments are analyzed. There is a corresponding good
relationship between outburst flow hydraulic characteristics and sandbars development
characteristics: the difference between the sediment concentration and the sediment
carrying capacity of the flow will determine the erosion and accumulation of sediments
that affect sandbars developments. The sandbars developments are an intuitive
manifestation of the changes in outburst flow hydraulic characteristics.

**Author contribution**

Xiangang Jiang was responsible for the experiments, article thinking, and writing.
Haiguang Cheng was responsible for calculating the article parameters. Lei Gao was
responsible for the article's pictures, and Weiming Liu was responsible for checking the
full article.



**Competing interests**

The authors declare that they have no known competing financial interests or personal relationships that could have appeared to influence the work reported in this paper.

**Acknowledgments**

This research has been supported by The National Natural Science Foundation of China (No. 41807289) and Key Laboratory of Ministry of Education for Geomechanics and Embankment Engineering, Hohai University (No. 202020) and Open fund of Key Laboratory of mountain hazards and surface processes, Chinese Academy of Sciences (No. KLMHESP-20-05).

**Code and data availability statement**

The codes and data that support the findings of this study are available from the corresponding author upon reasonable request.

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
