# Peer review of "The formation and geometry characteristics of boulder bars due to"

_Earth Surface Dynamics, 2020_

## Referee Comment (RC1) · Anonymous Referee #1 · 9 Jan 2021

General Comments: The manuscript by Dr. Jiang and colleagues summarizes results of an experiment investigating bar dynamics following breach of a landslide dam. The manuscript appears to be a re-working of results from a similar paper published by the same lead author in 2020 in the journal 'Landslides' (Jiang et al., 2020, cited in the manuscript). The experimental design appears sound, the experiment is well documented, and the results appear different enough from that paper to justify a separate publication. Nonetheless, the current manuscript suffers from a confusion of terminology and formative processes of the primary sedimentary body being investigated (fluvial bars), is lacking in scientific justification, and does not effectively communicate the novel scientific contribution of the experimental results. It is my judgement that

the results of the experiment could make a contribution to the scientific community, but the manuscript needs very substantial revision to meet the aims and scope of Earth Surface Dynamics. The other recommendation would be to revise the manuscript and submit to journal with different aims and scope.

Specific Comments:

1. The use of the term 'sandbar' is ill-founded. The experiment does, in fact, use a substrate that is approximately 40 percent sand. However, because of the scale and high Froude conditions (>2), the experiment best represents a canyon, gravel bed system. The formative processes of 'sandbars' in this experimental design are entirely different than the sedimentary bodies described in lines 54 to 108 of the Introduction. In that section, there is extensive review of sedimentary bodies that are not genetically nor stratigraphically related to the sandbodies formed in these experiments which, at the scale of the experiment are gravel alternate bars. The fact that the bars in the experiment migrate in the upstream direction is evidence that the experiments are simulating Froude-supercritical (diffusive) conditions (Shaw and McElroy, 2016), whereas most of the sandbars described in the Introduction (except those formed by landslide dams) are formed by translative depositional processes. I would suggest the authors re-visit the process scaling of the experiments to re-frame and strengthen the experimental justification and basis, and the scientific contributions of the results. Kleinhans et al. (2014) and Shaw and McElroy (2016) provide excellent discussions on linkages between sedimentary processes in flumes and those in rivers.

2. The authors do not provide a clear basis and justification for the experiments. Neither a hypothesis nor scientific question are presented in the introductory material as a basis for the experiments. Instead, the justification appears to be that 'sandbars are important'. Because the authors appear to have confused sandbars in low-slope, low Froude-number rivers with gravel bars from outburst floods, this justification is moot. In line 52 of the Introduction, the author's state "Sandbars are one typical landform formed during the outburst flood evolution (Turzewski et al., 2019; Jiang and Wei, 2020; Wu

et al., 2020).” Neither the Turzewski nor Wu papers describe sandbars at all, they describe gravel bars from outburst floods. Only the paper written by Dr. Jiang, which also appears to have confused sandbars with gravel bars, uses the term 'sandbars'. The authors should re-visit their results and the literature to provide the reader with a clear justification for the experiments by clearly stating a hypothesis or scientific question being addressed.

3. The manuscript lacks a clear description or discussion of the scientific contribution. The Results contain very long, detailed descriptions of the spatial-temporal dynamics of bar formation, geometries, and migration processes in the experiments. These descriptions could be shortened, and the scientific community would be better served with a discussion detailing how the results add to our understanding of bar formation from landslide outburst floods. For example, are the final geometries and along-stream scaling of the bars helpful in geologic interpretation of ancient bar deposits? Can they be used to improve interpretation of return frequency of certain outburst floods over recent geologic history? This manuscript simply does not contain any discussion linking the experimental results to the broader scientific literature, nor does it effectively relay the importance of the results to interpretation or prediction of landslide-dam outburst events.

References:

Kleinhans, M.G., van Dijk, W.M., van de Lageweg, W.I., Hoyal, D.C., Markies, H., van Maarseveen, M., Roosendaal, C., van Weesep, W., van Breemen, D., Hoendervoogt, R. and Cheshier, N., 2014. Quantifiable effectiveness of experimental scaling of river- and delta morphodynamics and stratigraphy. Earth-Science Reviews, 133, pp.43-61.

Shaw, J.B. and McElroy, B., 2016. Backwater number scaling of alluvial bed forms. Journal of Geophysical Research: Earth Surface, 121(8), pp.1436-1455.

---

## Referee Comment (RC2) · Anonymous Referee #2 · 14 Jan 2021

**Summary:**

The authors use a flume study to understand the effects of outburst flooding on downstream sandbar development. Different dam geometries (width and downstream slope angle) and a constant dam height were used. The upstream pool was allowed to fill and then overtop and fail under the same constant flow rate in all experiments. The authors relate bar frequency and volume to different dam geometries, and also note that bars tend to grow upstream during the experiments. The authors proceed to relate their observations to the flow hydraulics and sediment concentrations during the experiment. While the experimental set up seems reasonable, and the general result

reproducible, there are several parts of the analysis that are flawed. For example, the "sandbars" are not scaled appropriately, and in fact the median grain size is gravel in the experiment. Instead, these grains are equivalent to very coarse (boulders?) grains in the field scale. The framing of the introduction and paper in general is therefore not appropriate. Further, the Froude numbers during these experiments are all supercritical, leading to spurious correlations between transport capacity and flow depth (for example, I assume that dimensionless shear stress is calculated using subcritical flow assumptions via the depth-slope product embedded in u\*). Nor is it clear how sediment concentration (figure 8) was calculated with the reference to Laursen given. Later, they use the Meyer Peter-Muller equation to calculate bedload, but, again, not considering the supercritical flow regime of the experiments and the influence on energy slope as far as I can deduce. Therefore, it is difficult to interpret whether any of the results in sections 4 and 5 are valid. I believe this paper should be rejected at this time – perhaps it could be resubmitted with sufficient re-analysis, but I would suggest the authors also consider a different journal.

**General Comments:**

Introduction: make it clear how the background information will provide context for the results of the study. For example, the reference to Demirci et al. (2014) does not provide much insight into how these results for a coastal beach will provide context for this study. The authors could use the introduction to describe more precisely how these different previous studies relate to sandbars formed in settings 1, 2, and 3 described on lines 67-70. And then state how the sandbars in this study fit within one of those settings, or whether they are some different phenomenon related to outburst floods (as is implied). Further, the references should be more directly related to the coarse-grained alternate bars that form during the experiment, rather than sandbars.

Section 3: It would be very useful to have some information on grain size on the bars in this section. Much of the sediment in the experiment seems equivalent to boulders in the field case, and the coarse sediment seems to comprise much of the bar material.

**ESurfD**
Even in that case, the grain size of the sediment is going to be a very important factors in depositional patterns and should also be reported.

Section 4: It is confusing that sediment concentration is calculated using a reference to Laursen, with no reference to how this was done or whether we are talking about bed load or suspended load. For true sandbars, it seems that the suspended sediment component would dominate. Later in this section, the MPM formula is used for bed load transport capacity, but how is u\* defined given the supercritical flow conditions. I don't know if it is appropriate to use MPM without consideration of the effect of Froude number on the energy slope; this may lead to spurious negative correlations between Froude number and transport capacity.

Section 5: given the unknown equation to calculate sediment concentration, and uncertainty in the calculation of sediment transport capacity described above, I don't know how to interpret the results of this section.

Line Comments:

29: It "is" found...

35: Exchange "reference the research on" with "can be applied to"

41: delete "collapses" and just use "landslides"

60: "At present, much research..."

74: throughout the introduction, I suggest replacing semicolons with periods and starting new sentences.

118-131: good concluding paragraph of the introduction

156: spelling: "gravels"

156-157: Was there any dimensional scaling of the grain size? What would this sediment size correspond to in a field setting?
180: Were the balls buoyant in the flow?

188: Are you only able to measure the height along the flume wall? Rather than the average height across the channel?

220: Are these sandbars or gravel bars? They look to be dominated by the coarse fraction in the photos.

227-229: This sounds like alternate bar formation, for which there is significant literature that was not discussed in the introduction.

314-316: I don't understand why a smaller discharge would lead to a larger bar spacing. Please elaborate.

297-324: It would be useful to have a table or figure to show these differences between the experiments explicitly, or discuss in the context of Figure 5.

401: Please provide some more on the calculation based on Laursen; bed load? Suspended load?

407-422; Figure 8: I have no basis to judge any of this section because I do not know how the authors calculated these values with the available data. Using the surface velocity in different sections as measured with ball movement? What was the grain size used in the concentration calculation?

422: Spelling: abdomens? I think a different word was intended.

444: Were the concentration calculations using Laursen based on bedload as well?

451: equation 4 is not correct – need to square u\* in the numerator

473-477: Not sure I agree with this logic. These Froude numbers are well over 1 in the supercritical regime. The shear stress as calculated is lower at higher Froude numbers because it will be shallower, but the velocity will actually be even greater.

Figure 3: It looks like these are essentially alternate bars forming in a straight flume
channel – you state that this is similar to the field setting, but are the bar locations sometimes also controlled by the presence of obstructions?

Figure 4: I wonder if this figure could be simplified to focus on the key points in the discussion of the figure that describe the 3 variations of response.

Figure 5: This is a complex of a figure relative to its discussion in the text; the scale bar doesn't allow us to see much of a trend except for length. There is not consistency in the labeling scheme (dots for length, triangles for width, for example; same colors for the same model runs).

Figure 6: Same comment as figure 5 with regard

**ESurfD**

---

## Author Comment (AC1) · 10 Apr 2021

Response to the Reviewer 1 General Comments: The manuscript by Dr. Jiang and colleagues summarizes results of an experiment investigating bar dynamics following breach of a landslide dam. The manuscript appears to be a re-working of results from a similar paper published by the same lead author in 2020 in the journal 'Landslides' (Jiang et al., 2020, cited in the manuscript). The experimental design appears sound, the experiment is well documented, and the results appear different enough from that paper to justify a separate publication. Nonetheless, the current manuscript suffers from a confusion of terminology and formative processes of the primary sedimentary

body being investigated (fluvial bars), is lacking in scientific justification, and does not effectively communicate the novel scientific contribution of the experimental results. It is my judgement that the results of the experiment could make a contribution to the scientific community, but the manuscript needs very substantial revision to meet the aims and scope of Earth Surface Dynamics.

Thanks a lot for the reviewer's comments. In the revised manuscript, we have pointed out this manuscript's contributions to the scientific community: "Therefore, results in this paper can be applied to the river channel's geomorphological characteristics analysis triggered by overtopped landslide dam failure". Please see lines 31-33 in the revised manuscript. And we re-examined the results of the experiment and confirmed that the boulder bar was formed in the experiment instead of the sandbar. We also have made other revisions to the manuscript based on the reviewers' comments. Please see the revision.

Specific Comments: 1.The use of the term 'sandbar' is ill-founded. The experiment does, in fact, use a substrate that is approximately 40 percent sand. However, because of the scale and high Froude conditions (>2), the experiment best represents a canyon, gravel bed system. The formative processes of 'sandbars' in this experimental design are entirely different than the sedimentary bodies described in lines 54 to 108 of the Introduction. In that section, there is extensive review of sedimentary bodies that are not genetically nor stratigraphically related to the sandbodies formed in these experiments which, at the scale of the experiment are gravel alternate bars. The fact that the bars in the experiment migrate in the upstream direction is evidence that the experiments are simulating Froude-supercritical (diffusive) conditions (Shaw and McElroy, 2016), whereas most of the sandbars described in the Introduction (except those formed by landslide dams) are formed by translative depositional processes. I would suggest the authors re-visit the process scaling of the experiments to re-frame and strengthen the experimental justification and basis, and the scientific contributions of the results. Kleinhans et al. (2014) and Shaw and McElroy (2016) provide excellent

discussions on linkages between sedimentary processes in flumes and those in rivers.

Thanks a lot for the reviewer's comments. The authors have discussed this comment and agreed with the reviewer. We reviewed the experimental screen and confirmed that the bars formed in the experiment were boulder bars, which correspond to the boulder bars in the field (Wu et al., 2020; Turzewski et al., 2019). We have corrected the term "sandbar" to "boulder bar". The Introduction has been rewritten focused on boulder bars, the literatures have been recited, and lines 54-108 of the introduction of the original article have been revised by us to ensure accurate description of the boulder bar. The rewritten Introduction is as follows: "Activities such as rainfalls and earthquakes often cause landslides, which block the river to form a water-retaining body similar to a reservoir dam, called a landslide dam (Takahashi, 2007; Costa and Schuster, 1988; Casagli, 2003). According to statistics, 85 % of the dams failed within one year after formations, and more than 50 % of the dams breach with overtopping mode (Costa and Schuster, 1988). When the dam breach, the storage water erupt and flow to the downstream riverbed. Many studies on the influence of flood geomorphology and sedimentary characteristics have proved that the outburst flood energy is huge, and it can entrain and transport materials of various sizes, from clay to boulders. A large number of boulders gather in the river to form bars, namely boulder bars. The downstream riverbed's geomorphology will be significantly affected and undergo significant changes (Lamb and Fonstad, 2010; Maizels, 1997; Russell and Knudsen, 1999; Marren and Schuh, 2009; Benito and O'Connor, 2003; Carling, 2013; Wu et al., 2020). Boulder bars are one common landform formed during the outburst flood evolution (Turzewski et al., 2019; Jiang and Wei, 2020; Wu et al., 2020). For example, in the 2000 year, Yigong outburst flood, due to its huge lake storage, formed many huge boulder bars on the river bed. The boulder bars had a significant impact on the development of the river. And Wu et al. (2020) investigated the impact of this event on river morphology and analyzed the shapes and geometric characteristics of the boulder bars caused by the overtopping flood. And they found that the boulder bar components are poorly sorted. Turzewski et al. (2019) studied the particle gradation of the boulder

bars during the Yigong River landslide dam failure process. They found that the boulder bars' particle sizes decrease along the lower reaches of the river bed. But they did not analyze the evolution characteristics of boulder bar's size in detail. Lamb and Fonstad (2010) suggested that the rising and falling stages of the outburst flood had a greater impact on riverbed geomorphology and analyzed the characteristics of the median diameter of material in boulder bar. Because lack of field investigations about the growth characteristics of boulder bars during the landslide dam failure process in the field, some researchers had conducted landslide dam failure experiments in the lab (Jiang and Wei, 2020; Ashworth, 1996). Ashworth (1996) used flume experiments to study the boulder bar's growth. However, in their experiment, the inflow conditions are quite different from the outburst flood. Therefore, the research results' applicability to the boulder bar formed by the outburst flood remains uncertain. Jiang and Wei (2020) qualitatively analyzed the formation process of boulder bar in the evolution of overtopping outburst floods using dam failure experiments and initially discussed the characteristics of geometric dimensions of boulder bars after dam failure. However, the characteristics of the boulder bar's positions and geometric sizes during the dam failure process have not been analyzed. Above all, no matter whether it is field observations or indoor experiments, the boulder bar's development characteristic during the landslide dam overtopping failure process has not been proved. This paper focuses on the formation processes, the geometrical size characteristics of boulder bars in the downstream channel during the overtopping failure process, and how the dam volume and the released flood volume affect boulder bars' total volume. Firstly, through flume experiments, boulder bars' formation processes on the downstream channel under the dammed lake failure condition were reproduced. Then, based on the experimental data, the development characteristics of boulder bars' upstream and downstream edges were analyzed. Furthermore, statistical analysis of boulder bars geometrical dimensions at each moment during the failure process, such as length, width, height, and volume, had been carried out to obtain boulder bars' size characteristics. And then, by analyzing the total volume of the boulder bar under different dam volumes and

the released flood volumes, the influences of the released flood volume and dam volume on the boulder bar total volume were obtained. Finally, compare the boulder bar formed by the Yigong outburst flood and the boulder bar formed by the experiment to verify this experiment's reliability." In addition, as the reviewer said, the boulder bar in the experiment is indeed formed by the translational deposition of bedload. As shown in Figure 1, when the discharge is reduced, some gravel will stay on the river bed and hinder the advancement of the upstream flow, and part of the sediment in the flow will be deposited. As time goes by, the accumulation of sediment on the side of the boulder bar increases, and the boulder bar appears to develop upstream. We have added the explanations as the reviewer's suggestion in section 3.2. The revised section 3.2 is given in the response to the next reviewer's comment.

Figure 1. boulder bar obstructs the outburst flow to the river bed lower reaches

Reference: Wu C.H., Hu, K.H., Liu, W.M., Wang, H., Hu, X.D., and Zhang, X.P.: Morpho-sedimentary and stratigraphic characteristics of the 2000 Yigong River landslide dam outburst flood deposits, eastern Tibetan Plateau, Geomorphology, 107293, https://doi.org/10.1016/j.geomorph.2020.107293, 2020. Turzewski, M.D., Huntington, K.W., and Leveque, R.J.: The Geomorphic Impact of Outburst Floods: Integrating Observations and Numerical Simulations of the 2000 Yigong Flood, Eastern Himalaya, Journal of Geophysical Research: Earth Surface, 124, 5, https://doi.org/10.1029/2018JF004778, 2019.

2. The authors do not provide a clear basis and justification for the experiments. Neither a hypothesis nor scientific question are presented in the introductory material as a basis for the experiments. Instead, the justification appears to be that 'sandbars are important'. Because the authors appear to have confused sandbars in low-slope, low Froude number rivers with gravel bars from outburst floods, this justification is moot. In line 52 of the Introduction, the author's state "Sandbars are one typical landform formed during the outburst flood evolution (Turzewski et al., 2019; Jiang and Wei, 2020; Wu et al., 2020)." Neither the T urzewski nor Wu papers describe sandbars at all, they describe gravel bars from outburst floods. Only the paper written by Dr. Jiang, which also appears to have confused sandbars with gravel bars, uses the term 'sandbars'. The authors should revisit their results and the literature to provide the reader with a clear justification for the experiments by clearly stating a hypothesis or scientific question addressed

Thank the reviewer very much for the comments. We fully agree with the reviewer's opinion. We have corrected "sandbar" to "boulder bar" in the revision. It was noticed that most researchers focused on distribution characteristics and consisted material characteristics of boulder bars triggered by landslide dam overtopped failure after the dam failure based on field investigations (Wu et al., 2020; Turzewski et al., 2019). However, the boulder bar' formation process and development characteristic during the process of dam failure are still not clear. Therefore, we have proved the formation and development of the boulder bar during the failure process of the landslide dam through the flume experiment. We have pointed out the scientific question and rewritten the introduction section in the revision.

Reference: Wu C.H., Hu, K.H., Liu, W.M., Wang, H., Hu, X.D., and Zhang, X.P.: Morpho-sedimentary and stratigraphic characteristics of the 2000 Yigong River landslide dam outburst flood deposits, eastern Tibetan Plateau, Geomorphology, 107293, https://doi.org/10.1016/j.geomorph.2020.107293, 2020. Turzewski, M.D., Huntington, K.W., and Leveque, R.J.: The Geomorphic Impact of Outburst Floods: Integrating Observations and Numerical Simulations of the 2000 Yigong Flood, Eastern Himalaya, Journal of Geophysical Research: Earth Surface, 124, 5, https://doi.org/10.1029/2018JF004778, 2019.

3. The manuscript lacks a clear description or discussion of the scientific contribution. The Results contain very long, detailed descriptions of the spatial-temporal dynamics of bar formation, geometries, and migration processes in the experiments. These descriptions could be shortened, and the scientific community would be better served with a discussion detailing how the results add to our understanding of bar formation

from landslide outburst floods. For example, are the final geometries and along-stream scaling of the bars helpful in geologic interpretation of ancient bar deposits? Can they be used to improve interpretation of return frequency of certain outburst floods over recent geologic history? This manuscript simply does not contain any discussion linking the experimental results to the broader scientific literature, nor does it effectively relay the importance of the results to interpretation or prediction of landslide-dam outburst events.

Thanks a lot for the reviewer's comments. We have simplified the description of the position and size characteristics of the boulder bar in the experiment according to the reviewer's suggestions. The simplified description is as followsïijŽ "3. 
[revised manuscript text omitted]
." We study the formation process and growth characteristic of the boulder bar during the landslide dam overtopping failure process. The boulder bar's position and the change characteristics of geometric dimensions in the process of dam failure were carefully discussed. In addition, we have added a new discussion section to compare the experimental results with a field case (Wu et al., 2020; Turzewski et al., 2019). The experimental results and field data are consistent, indicating that the experimental results can provide references for the study of the formation and growth of the boulder bar formed by the outburst flood. The Discussion section is as follows: " 5. Discussion The field data of the Yigong landslide dam are used to verify the reliability of the results in this paper. Turzewski et al. (2019) investigated the boulder bars in the Yigong River triggered by the Yigong landslide dam outburst flood in 2000. They found that the number of boulder bars is about 0.69 to 0.77 times the ratio of river bed length to dam length for the boulder bar frequent region. In this study, boulder bars were distributed in the 8 m length of the channel, which is 4 to 7 times of dam length. It reflected the number of boulder bars was 0.4 to 1.0 times the ratio of river bed length to dam length. By comparing the experimental data and the field data of Turzewski et al. (2019), it can be found that field data falls within the range of experimental data. Experimental models took more influence factors into account in this paper, while the field data of Turzewski et al. (2019) only focused on the Yigong landslide dam case. This may be why the field data range is smaller than the experimental data in this paper. Wu et al. (2020) classified the boulder bars in the downstream reaches of the Yigong River into three types according to their shapes and used the length to width ratio as the indicator of a bar shape. The 16 boulder bars in the downstream reaches of the

Yigong River have a length to width ratio of 2.5-15. As can be seen from Fig. 9, the length to width ratio of the boulder bar formed in this experiment is in the range of 7 to 16, which indicates the field data could prove the experimental results.

Figure. 9 The ratio of boulder bar length to width Turzewski et al. (2019) measured the sizes of boulder bars. They found that grain sizes of boulder bars decrease downstream. In this experiment, boulder bar materials from different river bed sections were collected. And after screening and analysis, it was found that as the distance between the boulder bar and the dam increases, the particle diameter in the bars shows a decreasing trend, as shown in Fig. 10. This feature is consistent with the description of Turzewski et al. (2019).

Figure. 10 Gradation curve of the boulder bar materials. Notation: U, M, D, MU, and MD, represent the boulder bar near the upstream reaches, the boulder bar near the middle reaches, the boulder bar near the downstream reaches, the boulder bar near the middle-upstream reaches, and the boulder bar near the middle-downstream reaches, respectively. Based on the above points, it can be seen that the experimental results in this paper are consistent with the actual boulder bars in the field. Therefore, the experimental results can provide guidance for the field study of the boulder bar formed by the outburst flood."

Reference: Wu C.H., Hu, K.H., Liu, W.M., Wang, H., Hu, X.D., and Zhang, X.P.: Morpho-sedimentary and stratigraphic characteristics of the 2000 Yigong River landslide dam outburst flood deposits, eastern Tibetan Plateau, Geomorphology, 107293, https://doi.org/10.1016/j.geomorph.2020.107293, 2020. Turzewski, M.D., Huntington, K.W., and Leveque, R.J.: The Geomorphic Impact of Outburst Floods: Integrating Observations and Numerical Simulations of the 2000 Yigong Flood, Eastern Himalaya, Journal of Geophysical Research: Earth Surface, 124, 5, https://doi.org/10.1029/2018JF004778, 2019.

Please also note the supplement to this comment:

https://esurf.copernicus.org/preprints/esurf-2020-92/esurf-2020-92-AC1-supplement.pdf

[Figure]

**Figure 1**. boulder bar obstructs the outburst flow to the river bed lower reaches

[Figure]

**Figure. 3** The riverbed morphology at six different moments during the boulder bars' formations and growths process for the T7 experiment. The boulder bars in the figure are marked with blue dotted lines.

**Fig. 1.**

[Figure]

**Fig. 2.**

[Figure]

**Fig. 3.**

**Figure. 4** The boulder bars' locations during the dam failure process. Notation: (a) to (h) represent the boulder bars' locations for T1-T8 tests, respectively. The red lines in the figure represent the boulder bars, and the orange rectangles represent the channels. The numbers at both ends of the red lines represent the distances between the upstream and downstream edges of boulder bars and the dam toe.

[Figure]

**Figure. 5** The lengths, widths, and heights of the boulder bars: (a) sizes of the boulder bars near the upstream reaches; (b) sizes of the boulder bars near the middle reaches; (c) sizes of the boulder bars near the downstream reaches. Notation: L, W, and H represent the length, width, and height of the boulder bar, respectively. i represents the Ti experiment. For example, MUL6 indicates the length of the boulder bar near the middle-upstream reaches for the T1 test.

**Fig. 4.**

[Figure]

**Figure. 6** Volumes of boulder bars. Notation: UVi, MVi, DVi, MUVi, MDVi represent the volume of the boulder bar near the upstream reaches,the boulder bar near the middle reaches, the boulder bar near the downstream reaches, the boulder bar near the middle-upstream reaches, and the boulder bar near the middle-downstream reaches, respectively. For example, UV1 means the volume of the boulder bar near the upstream reaches of the T1 test.

[Figure]

**Figure. 9** The ratio of boulder bar length to width

**Fig. 5.**

---

## Author Comment (AC2) · 10 Apr 2021

Response to the Reviewer 2 Summary: The authors use a flume study to understand the effects of outburst flooding on downstream sandbar development. Different dam geometries (width and downstream slope angle) and a constant dam height were used. The upstream pool was allowed to fill and then overtop and fail under the same constant flow rate in all experiments. The authors relate bar frequency and volume to different dam geometries, and also note that bars tend to grow upstream during the experiments. The authors proceed to relate their observations to the flow hydraulics and sediment concentrations during the experiment. While the experimental set up seems

reasonable, and the general result reproducible, there are several parts of the analysis that are flawed. For example, the "sandbars" are not scaled appropriately, and in fact the median grain size is gravel in the experiment. Instead, these grains are equivalent to very coarse (boulders?) grains in the field scale. The framing of the introduction and paper in general is therefore not appropriate. Further, the Froude numbers during these experiments are all supercritical, leading to spurious correlations between transport capacity and flow depth (for example, I assume that dimensionless shear stress is calculated using subcritical flow assumptions via the depth-slope product embedded in u*). Nor is it clear how sediment concentration (figure 8) was calculated with the reference to Laursen given. Later, they use the Meyer Peter-Muller equation to calculate bedload, but, again, not considering the supercritical flow regime of the experiments and the influence on energy slope as far as I can deduce. Therefore, it is difficult to interpret whether any of the results in sections 4 and 5 are valid.

Thanks a lot for the reviewer's comments. As the reviewer concerned, the bar formed in the experiment was composed of a lot of coarser materials, which should be named "boulder bar". We have corrected the term "sandbar" to "boulder bar" in the revision. Considering the content of the original manuscript is more, and the sections 4 and 5 have caused the reviewer greater confusion, we have deleted the sections 4 and 5 of the original manuscript after our careful consideration. And in the revised manuscript, the influence of the dam volume and the released flood volume on the growth of the boulder bar was added (section 4 of the revision), and the results of this experiment were compared with the Yigong flood in the Discussion section (section 5 of the revision), which proved the reliability of the results of this experiment. It shows that the research results of this paper can provide reference for the research on the formation and development of the boulder bar formed by the overtopping outburst flood. The revised sections 4 and 5 are as followsïijŽ "4. Influences of the dam volume and the released flood volume on total boulder bar volume The boulder bar's formation and development are inseparable from the combined action of outburst flow and sediment. The landslide dam can provide materials for the development of the boulder bar, while

[revised manuscript text omitted]

General Comment: Introduction: make it clear how the background information will provide context for the results of the study. For example, the reference to Demirci et al. (2014) does not provide much insight into how these results for a coastal beach will provide context for this study. The authors could use the introduction to describe more precisely how these different previous studies relate to sandbars formed in settings 1, 2, and 3 described on lines 67-70. And then state how the sandbars in this study fit within one of those settings, or whether they are some different phenomenon related to outburst floods (as is implied). Further, the references should be more directly related to the coarse-grained alternate bars that form during the experiment, rather than sandbars.

Thanks a lot for your comments. According to your valuable suggestion, we have corrected the term "sandbar" to "boulder bar" in the revised manuscript. Also, we have rewritten the introduction part of the article and recited references related to the boulder bar to ensure the correct citation of the references. The rewritten Introduction is as follows: "Activities such as rainfalls and earthquakes often cause landslides, which block the river to form a water-retaining body similar to a reservoir dam, called a landslide dam (Takahashi, 2007; Costa and Schuster, 1988; Casagli, 2003). According to statistics, 85 % of the dams failed within one year after formations, and more than 50 % of the dams breach with overtopping mode (Costa and Schuster, 1988). When the dam breach, the storage water erupt and flow to the downstream riverbed. Many studies on the influence of flood geomorphology and sedimentary characteristics have proved that the outburst flood energy is huge, and it can entrain and transport materials of various sizes, from clay to boulders. A large number of boulders gather in the river to form bars, namely boulder bars. The downstream riverbed's geomorphology will be significantly affected and undergo significant changes (Lamb and Fonstad, 2010; Maizels, 1997; Russell and Knudsen, 1999; Marren and Schuh, 2009; Benito and O'Connor, 2003; Carling, 2013; Wu et al., 2020). Boulder bars are one common landform formed during the outburst flood evolution (Turzewski et al., 2019; Jiang and Wei, 2020; Wu et al., 2020). For example, in the 2000 year, Yigong outburst flood, due

to its huge lake storage, formed many huge boulder bars on the river bed. The boulder bars had a significant impact on the development of the river. And Wu et al. (2020) investigated the impact of this event on river morphology and analyzed the shapes and geometric characteristics of the boulder bars caused by the overtopping flood. And they found that the boulder bar components are poorly sorted. Turzewski et al. (2019) studied the particle gradation of the boulder bars during the Yigong River landslide dam failure process. They found that the boulder bars' particle sizes decrease along the lower reaches of the river bed. But they did not analyze the evolution characteristics of boulder bar's size in detail. Lamb and Fonstad (2010) suggested that the rising and falling stages of the outburst flood had a greater impact on riverbed geomorphology and analyzed the characteristics of the median diameter of material in boulder bar. Because lack of field investigations about the growth characteristics of boulder bars during the landslide dam failure process in the field, some researchers had conducted landslide dam failure experiments in the lab (Jiang and Wei, 2020; Ashworth, 1996). Ashworth (1996) used flume experiments to study the boulder bar's growth. However, in their experiment, the inflow conditions are quite different from the outburst flood. Therefore, the research results' applicability to the boulder bar formed by the outburst flood remains uncertain. Jiang and Wei (2020) qualitatively analyzed the formation process of boulder bar in the evolution of overtopping outburst floods using dam failure experiments and initially discussed the characteristics of geometric dimensions of boulder bars after dam failure. However, the characteristics of the boulder bar's positions and geometric sizes during the dam failure process have not been analyzed. Above all, no matter whether it is field observations or indoor experiments, the boulder bar's development characteristic during the landslide dam overtopping failure process has not been proved. This paper focuses on the formation processes, the geometrical size characteristics of boulder bars in the downstream channel during the overtopping failure process, and how the dam volume and the released flood volume affect boulder bars' total volume. Firstly, through flume experiments, boulder bars' formation processes on the downstream channel under the dammed lake failure condition were

reproduced. Then, based on the experimental data, the development characteristics of boulder bars' upstream and downstream edges were analyzed. Furthermore, statistical analysis of boulder bars geometrical dimensions at each moment during the failure process, such as length, width, height, and volume, had been carried out to obtain boulder bars' size characteristics. And then, by analyzing the total volume of the boulder bar under different dam volumes and the released flood volumes, the influences of the released flood volume and dam volume on the boulder bar total volume were obtained. Finally, compare the boulder bar formed by the Yigong outburst flood and the boulder bar formed by the experiment to verify this experiment's reliability."

Section 3: It would be very useful to have some information on grain size on the bars in this section. Much of the sediment in the experiment seems equivalent to boulders in the field case, and the coarse sediment seems to comprise much of the bar material. Even in that case, the grain size of the sediment is going to be a very important factors in depositional patterns and should also be reported.

We fully agree with the reviewer's opinion. We report the material size information of the boulder bar downstream of the river bed (section 5 in the revised manuscript). We found that the farther away from the dam, the smaller the median particle size of the boulder bar material for the boulder bar on the river bed. The experimental results are in good agreement with the Yigong flood (Turzewski et al., 2019). We have added the related descriptions and explanations in the 3th paragraph of section 5. The details are as followsïijŽ "Turzewski et al. (2019) measured the sizes of boulder bars. They found that grain sizes of boulder bars decrease downstream. In this experiment, boulder bar materials from different river bed sections were collected. And after screening and analysis, it was found that as the distance between the boulder bar and the dam increases, the particle diameter in the bars shows a decreasing trend, as shown in Fig. 10. This feature is consistent with the description of Turzewski et al. (2019).

Figure. 10 Gradation curve of the boulder bar materials. Notation: U, M, D, MU, and MD, represent the boulder bar near the upstream reaches, the boulder bar near

the middle reaches, the boulder bar near the downstream reaches, the boulder bar near the middle-upstream reaches, and the boulder bar near the middle-downstream reaches, respectively. "

Reference: Turzewski, M.D., Huntington, K.W., and Leveque, R.J.: The Geomorphic Impact of Outburst Floods: Integrating Observations and Numerical Simulations of the 2000 Yigong Flood, Eastern Himalaya, Journal of Geophysical Research: Earth Surface, 124, 5, https://doi.org/10.1029/2018JF004778, 2019.

Section 4: It is confusing that sediment concentration is calculated using a reference to Laursen, with no reference to how this was done or whether we are talking about bed load or suspended load. For true sandbars, it seems that the suspended sediment component would dominate. Later in this section, the MPM formula is used for bed load transport capacity, but how is u* defined given the supercritical flow conditions. I don't know if it is appropriate to use MPM without consideration of the effect of Froude number on the energy slope; this may lead to spurious negative correlations between Froude number and transport capacity.

Thanks a lot for the reviewer's comments. it is difficult to measure the concentration and sediment carrying capacity of outburst flow exactly and timely during experimental process. So, we adopted the calculated methods to obtain the concentration and sediment carrying capacity values in the first version manuscript. But the calculated values may be different with the experimental data. The content of section 4 in the original manuscript may cause confusion to the readers. Therefore, we decided to delete the content of section 4 in the original manuscript The section 4 of the revised draft is as follows: "4. Influences of the dam volume and the released flood volume on total boulder bar volume The boulder bar's formation and development are inseparable from the combined action of outburst flow and sediment. The landslide dam can provide materials for the development of the boulder bar, while the outburst flow provides hydraulic conditions. Figure. 7 shows the boulder bars' total volume on the river bed when the dam fully failed. It can be seen that the total volume of the boulder bars is

much lower than the dam volume. The volumes were about 0.079 to 0.127, 0.017 to 0.078, and 0.015 to 0.041 times of the initial dam volumes for the boulder bars near the upstream reaches, the boulder bars near the middle reaches, and the boulder bars near the downstream reaches, respectively. The ratio of the total volume of the boulder bars to the dam volume is 0.138 to 0.208. It shows that only a small part of the dam material participates in the boulder bar's formation and development. During the process, most of the dam material was taken away by the outburst flow. Moreover, when the dam volume decreases, the amount of sediment involved in the development of the boulder bar decreases. The total volume of the boulder bars on the river bed also shows a decreasing trend. This experiment counted the released flood volume during the dam failure process, as shown in Fig. 8. It could be seen that the released flood volume in the dam failure process of the T1 to T8 experiments decreased. According to Figs. 7 and 8, it could be found that with the decrease of the released flood volume, the total volume of boulder bars on the river bed shows a decreasing trend. When the released flood volume is small in the dam failure process, a small amount of flood is not enough to transport many dam materials to the downstream riverbed. There is less sediment on the riverbed, and the deposit that can participate in the boulder bar's growth is less. Therefore, the total volume of the boulder bars on the river bed at the moment of complete dam failure decreased with the decrease of the released flood volume.

Figure. 7 The boulder bar's total volume and dam volume Figure. 8 The volume of water released during the dam failure. "

Section 5: given the unknown equation to calculate sediment concentration, and uncertainty in the calculation of sediment transport capacity described above, I don't know how to interpret the results of this section.

Thanks a lot for the reviewer's comments. After our careful discussion, we decided to delete section 5 from the original manuscript. In the revised manuscript, we compared the experimental results with Yigong case and discussed the reliability of experimental

results in section 5. The section 5 in the revised manuscript is as follows: " 5. Discussion The field data of the Yigong landslide dam are used to verify the reliability of the results in this paper. Turzewski et al. (2019) investigated the boulder bars in the Yigong River triggered by the Yigong landslide dam outburst flood in 2000. They found that the number of boulder bars is about 0.69 to 0.77 times the ratio of river bed length to dam length for the boulder bar frequent region. In this study, boulder bars were distributed in the 8 m length of the channel, which is 4 to 7 times of dam length. It reflected the number of boulder bars was 0.4 to 1.0 times the ratio of river bed length to dam length. By comparing the experimental data and the field data of Turzewski et al. (2019), it can be found that field data falls within the range of experimental data. Experimental models took more influence factors into account in this paper, while the field data of Turzewski et al. (2019) only focused on the Yigong landslide dam case. This may be why the field data range is smaller than the experimental data in this paper. Wu et al. (2020) classified the boulder bars in the downstream reaches of the Yigong River into three types according to their shapes and used the length to width ratio as the indicator of a bar shape. The 16 boulder bars in the downstream reaches of the Yigong River have a length to width ratio of 2.5-15. As can be seen from Fig. 9, the length to width ratio of the boulder bar formed in this experiment is in the range of 7 to 16, which indicates the field data could prove the experimental results.

Figure. 9 The ratio of boulder bar length to width Turzewski et al. (2019) measured the sizes of boulder bars. They found that grain sizes of boulder bars decrease downstream. In this experiment, boulder bar materials from different river bed sections were collected. And after screening and analysis, it was found that as the distance between the boulder bar and the dam increases, the particle diameter in the bars shows a decreasing trend, as shown in Fig. 10. This feature is consistent with the description of Turzewski et al. (2019).

Figure. 10 Gradation curve of the boulder bar materials. Notation: U, M, D, MU, and MD, represent the boulder bar near the upstream reaches, the boulder bar near

the middle reaches, the boulder bar near the downstream reaches, the boulder bar near the middle-upstream reaches, and the boulder bar near the middle-downstream reaches, respectively. Based on the above points, it can be seen that the experimental results in this paper are consistent with the actual boulder bars in the field. Therefore, the experimental results can provide guidance for the field study of the boulder bar formed by the outburst flood."

Line Comments: 29: It "is" found. . .

Thanks for the reviewer's guidance. The line 29 in the original manuscript is related to the Froude number. In the revised manuscript, we have deleted the content related to the Froude number. Therefore, this sentence is no longer in the revised manuscript. 35: Exchange "reference the research on" with "can be applied to"

Thanks a lot for the reviewer's comments. We have made corrections in accordance with the reviewer's requirements. See the line 32 of the revised manuscript for details: "results in this paper can be applied to the river channel's geomorphological characteristics analysis triggered by overtopped landslide dam failure."

41: delete "collapses" and just use "landslides"

Thank the reviewer for this suggestion. We have made corrections in accordance with the reviewer's requirements. See line 38 of the revised manuscript for details: "Activities such as rainfalls and earthquakes often cause landslides, which block the river to form a water-retaining body similar to a reservoir dam, called a landslide dam (Takahashi, 2007; Costa and Schuster, 1988; Casagli, 2003). "

60: "At present, much research. . ."

Thanks a lot for the reviewer's comments. We have corrected the sentence in the revision.

74: throughout the introduction, I suggest replacing semicolons with periods and starting new sentences.

Thanks very much for the reviewer's comments. We have rewritten the introduction section in the revised manuscript based on your valuable suggestion.

118-131: good concluding paragraph of the introduction

Thanks very much for the reviewer's compliment. We used these sentences in our rewritten Introduction: "Above all, no matter whether it is field observations or indoor experiments, the boulder bar's development characteristic during the landslide dam overtopping failure process has not been proved. This paper focuses on the formation processes, the geometrical size characteristics of boulder bars in the downstream channel during the overtopping failure process, and how the dam volume and the released flood volume affect boulder bars' total volume. Firstly, through flume experiments, boulder bars' formation processes on the downstream channel under the dammed lake failure condition were reproduced. Then, based on the experimental data, the development characteristics of boulder bars' upstream and downstream edges were analyzed. Furthermore, statistical analysis of boulder bars geometrical dimensions at each moment during the failure process, such as length, width, height, and volume, had been carried out to obtain boulder bars' size characteristics. And then, by analyzing the total volume of the boulder bar under different dam volumes and the released flood volumes, the influences of the released flood volume and dam volume on the boulder bar total volume were obtained. Finally, compare the boulder bar formed by the Yigong outburst flood and the boulder bar formed by the experiment to verify this experiment's reliability. "

156: spelling: "gravels"

Thanks a lot for the reviewer's comment. We have revised the spelling of the "gravels" : "The dam materials used in this study were mixtures of sand and gravels, with a median particle size D50 of 3.8 mm. "

156-157: Was there any dimensional scaling of the grain size? What would this sediment size correspond to in a field setting?

Thanks a lot for your comments. Although, there was not any dimensional scaling of the grain size, the type of materials used in the tests was similar to the filed, and the materials could ensure the overtopped failure mode happen for the tests. Most of the barrier dams in the field are mixtures of fine particles and coarse particles, therefore, we selected mixtures of sand and gravels as the experimental materials. With reference to Vallejo and Mawby (2000) and Wan and Fell (2008), considering the limitations of the experimental flume space and the size of the dam model, the experimental material adopted a median particle size of 3.8mm mixtures of sand and gravels to improve the possibility of overtopping failure of dam model. Therefore, the selection of dam model and material in this experiment could meet the experimental requirements.

Reference: Vallejo L. E., Mawby R.: Porosity influence on the shear strength of granular material–clay mixtures, Engineering Geology, 58(2):125-136, https://doi.org/10.1016/S0013-7952(00)00051-X , 2000. Wan C. F., Fell R.: Assessing the Potential of Internal Instability and Suffusion in Embankment Dams and Their Foundations, Journal of Geotechnical and Geoenvironmental Engineering, 134(3):401-407, https://doi.org/10.1061/(ASCE)1090-0241(2008)134:3(401), 2008,

180: Were the balls buoyant in the flow?

We are honored to be able to answer the reviewer's confusion. The balls used in the experiment have a small mass and can float on the flow surface, and can be used to measure the flow velocity.

188: Are you only able to measure the height along the flume wall? Rather than the average height across the channel?

We are honored to answer the reviewer's questions. We could measure the height along the flume wall timely and conveniently during the dam failure process. Although, the height of other positions maybe different with the section along the flume wall, but the difference is small. So we selected the boulder bar's section along the flume wall as concerned positions of the boulder bar and measured the height of the boulder bar

of these positions. Because of the irregular shape of the boulder bar, the height of the boulder bar is different at different position, so we take the average height along the wall of the flume as the representative height value of the boulder bar.

220: Are these sandbars or gravel bars? They look to be dominated by the coarse fraction in the photos.

Thanks a lot for the reviewer's reminder that gravel bars were formed in the river bed during the dam failure in the experiment, which corresponds to the boulder bar formed by the outburst flood in the field. We have also corrected this concept in the revised manuscript.

227-229: This sounds like alternate bar formation, for which there is significant literature that was not discussed in the introduction. Thanks a lot for the reviewer's comment. As the reviewer mentioned, we have changed the concepts of "sandbar" to "boulder bar". In the revised manuscript we have rewritten the Introduction on "boulder bar".

314-316: I don't understand why a smaller discharge would lead to a larger bar spacing. Please elaborate. It is a great honor for us to explain this phenomenon to the reviewer. Maybe we did not describe it clearly that the "discharge" is "inflow discharge". We have clarified this term in the revision. Although the inflow discharge in the experiment is small, but the stored water volume behind the dam may be large. When the water volume in front of the dam is large enough, the landslide dam will be over-topped, and the dam will be failure very quickly. Water will be released in a short time, and the outburst discharge may be large (Jiang and Wei, 2020; Carrivick 2010; Jiang and Wei, 2018). We have deleted the confusing sentences. In order to facilitate the readers to understand the degree of amplification of the discharge, we also give the peak discharge of 8 sets of experiments (Figure 1).

Figure 1. The breaching discharge hydrographs for T1 to T8 tests

Reference: Carrivick, J. L.: Dam break–outburst flood propagation and transient hydraulics: a geosciences perspective. J Hydrol, 380(3–4):338–355, https://doi.org/10.1016/j.jhydrol.2009.11.009, 2010. Jiang, X. G. and Wei, Y. W.: Natural dam breaching due to overtopping: effects of initial soil moisture. Bull Eng Geol Environ 78, 4821–4831, https://doi.org/10.1007/s10064-018-01441-7, 2018. Jiang, X.G., and Wei, Y.W: Erosion characteristics of outburst floods on channel beds under the conditions of different natural dam downstream slope angles, Landslides, 1-12, https://doi.org/10.1007/s10346-020-01381-y, 2020.

297-324: It would be useful to have a table or figure to show these differences between the experiments explicitly, or discuss in the context of Figure 5.

Thanks a lot for the reviewer's advice. We have revised the figure as the reviewer's suggestion. With the new Figure 4 in the revised manuscript, it is easy to understand the characteristic of the position of the boulder bar.

401: Please provide some more on the calculation based on Laursen; bed load? Suspended load?

After careful consideration, we decided to delete the calculation of the concentration in the revised manuscript. There is no more content related to concentration in the revised manuscript.

407-422; Figure 8: I have no basis to judge any of this section because I do not know how the authors calculated these values with the available data. Using the surface velocity in different sections as measured with ball movement? What was the grain size used in the concentration calculation?

After careful consideration, we decided to delete the calculation of the concentration in the revised manuscript. There are no more contents related to concentration in the revised manuscript.

422: Spelling: abdomens? I think a different word was intended. Thanks a lot for the

reviewer's suggestion. We replaced "abdomens" with "waists" in the revised manuscript (see line 259 in the revised manuscript).

444: Were the concentration calculations using Laursen based on bedload as well?

Thank you very much for the comment. After careful consideration, we decided to delete the calculation of the concentration in the revised manuscript. There are no more contents related to concentration in the revised manuscript.

451: equation 4 is not correct – need to square u* in the numerator

Thanks a lot for the reviewer's correction. We have deleted equation 4 in the revised manuscript.

473-477: Not sure I agree with this logic. These Froude numbers are well over 1 in the supercritical regime. The shear stress as calculated is lower at higher Froude numbers because it will be shallower, but the velocity will actually be even greater.

Thanks a lot for the reviewer's comment. This comment is very helpful to us. We all agree with the reviewer's point of view. After our discussion, we have deleted this part in the revised manuscript.

Figure 3: It looks like these are essentially alternate bars forming in a straight flume channel – you state that this is similar to the field setting, but are the bar locations sometimes also controlled by the presence of obstructions?

Thanks a lot for the reviewer's comment. The question raised by the reviewer is very valuable. Boulder bar locations are sometimes controlled by the presence of obstructions. The river bed downstream terrain conditions of the field landslide dam are more complicated. In the lab, we simplified the experimental conditions. For example we simplified the channel shape and omitted the obstacle in the channel. The straight channel used in the tests is a common simplified model (Jiang and Wei, 2020; Chen et al., 2015). It is convenient for us to use the straight channel model to summarize the characteristics of boulder bar's formation and development. When the river bed in the

experiment is not of equal width and straight, it is not conducive to drawing a general rule. In addition, the experimental dam model designed according to the method of Zhou et al., (2019) can reflect the actual characteristics of the field landslide dam. And according to the research of Vallejo and Mawby (2000) and Wan and Fell (2008), the materials for this experiment can be meet the experimental requirements.

Reference: Chen, S. C., Lin, T. W., and Chen, C. Y.: Modeling of natural dam failure modes and downstream riverbed morphological changes with different dam materials in a flume test, Engineering Geology, 188, 148-158, https://doi.org/10.1016/j.enggeo.2015.01.016, 2015. Jiang, X. G., and Wei, Y. W: Erosion characteristics of outburst floods on channel beds under the conditions of different natural dam downstream slope angles, Landslides, 1-12, https://doi.org/10.1007/s10346-020-01381-y, 2020. Vallejo L. E., Mawby R.: Porosity influence on the shear strength of granular material–clay mixtures, Engineering Geology, 58(2):125-136, https://doi.org/10.1016/S0013-7952(00)00051-X , 2000. Wan C. F., Fell R.: Assessing the Potential of Internal Instability and Suffusion in Embankment Dams and Their Foundations, Journal of Geotechnical and Geoenvironmental Engineering, 134(3):401-407, https://doi.org/10.1061/(ASCE)1090-0241(2008)134:3(401), 2008, Zhou, G. G. D., Zhou, M. J., Shrestha, M. S., Song, D. R., Choi, C. E., Cui, K. F. E., Peng, M., Shi, Z. M., Zhu, X. H., and Chen, H. Y.: Experimental investigation on the longitudinal evolution of landslide dam breaching and outburst floods, Geomorphology, 334, 29-43, https://doi.org/10.1016/j.geomorph.2019.02.035, 2019.

Figure 4: I wonder if this figure could be simplified to focus on the key points in the discussion of the figure that describe the 3 variations of response.

Thanks a lot for the reviewer's comment. Figure 4 in the manuscript shows the change of the position of the boulder bar during the dam failure process. Figure 4 can very intuitively and vividly express the change of the position of the boulder bar over time. Therefore, if possible, we would like to retain the contents of Figure 4. The revised Figure 4 is as follows:

(a)

(b)

(c)

(d)

(e)

(f)

(g)

(h) Figure. 4 The boulder bars' locations during the dam failure process. Notation: (a) to (h) represent the boulder bars' locations for T1-T8 tests, respectively. The red lines in the figure represent the boulder bars, and the orange rectangles represent the channels. The numbers at both ends of the red lines represent the distances between the upstream and downstream edges of boulder bars and the dam toe.

Figure 5: This is a complex of a figure relative to its discussion in the text; the scale bar doesn't allow us to see much of a trend except for length. There is not consistency in the labeling scheme (dots for length, triangles for width, for example; same colors for the same model runs).

Thanks a lot for the reviewer's comment. According to the reviewer's suggestion, we have modified Figure 5 in the manuscript. We hope that the revised figure can satisfy the reviewer. The revised figure is in the following:

Figure. 5 The lengths, widths, and heights of the boulder bars: (a) sizes of the boulder bars near the upstream reaches; (b) sizes of the boulder bars near the middle reaches; (c) sizes of the boulder bars near the downstream reaches. Notation: L, W, and H represent the length, width, and height of the boulder bar, respectively. i represents the Ti experiment. For example, MUL6 indicates the length of the boulder bar near the middle-upstream reaches for the T1 test.

Figure 6: Same comment as figure 5 with regard Thanks a lot for the reviewer's comment. This suggestion is very helpful to us. We have modified Figure 6 in the manuscript as suggested by the reviewer, as shown in the figure below

Figure. 6 Volumes of boulder bars. Notation: UVi, MVi, DVi, MUVi, MDVi represent the volume of the boulder bar near the upstream reachesïijŇthe boulder bar near the middle reaches, the boulder bar near the downstream reaches, the boulder bar near the middle-upstream reaches, and the boulder bar near the middle-downstream reaches, respectively. For example, UV1 means the boulder bar's volume near the dam toe of the T1 test

Please also note the supplement to this comment:
https://esurf.copernicus.org/preprints/esurf-2020-92/esurf-2020-92-AC2-supplement.pdf
* * *
[Figure]

Figure. 7 The boulder bar's total volume and dam volume

Figure. 8 The volume of water released during the dam failure.

Figure. 9 The ratio of boulder bar length to width

**Fig. 1.**

[Figure]

**Figure. 10** Gradation curve of the boulder bar materials. Notation: U, M, D, MU, and MD, represent the boulder bar near the upstream reaches, the boulder bar near the middle reaches, the boulder bar near the downstream reaches, the boulder bar near the middle-upstream reaches, and the boulder bar near the middle-downstream reaches, respectively.

[Figure]

Figure 1. The breaching discharge hydrographs for T1 to T8 tests

**Fig. 2.**

---

## Author Response (AR2)

**Responses to editor and reviewers**

Dear editor,

Thanks very much for taking your time to review this manuscript entitled "The formation processes and development characteristics of boulder bars due to outburst flood triggered by the overtopped landslide dam failure" (Esurf-2020-92). I appreciate all your comments and suggestions! Those comments are constructive for us to revise and improve our paper. We have read these comments carefully and tried our best to revise and improve the manuscript. The revised contents have been highlighted in blue in the revision. The followings are the point-to-point response for each comment. We hope our revision will make the manuscript acceptable for publication. Thanks again!

Best regards,

Xiangang Jiang

*Comments of Editor:*

*• points out that the manuscript is lacking a scientific justification of the study as well as an explanation of the contribution of the results to the scientific community. Your response states that you have addressed this in one sentence in the abstract. However, this text is needed Introduction, and the Discussion needs to more clearly explain how the results contribute to a wider understanding of morphologic changes following outburst floods.*

Thank you for this suggestion. We have added scientific justification for this research in the Introduction. In the Discussion section, the geometric size and distribution related data of 38 field boulder bars were supplemented. The boulder bars formed in this experiment were carefully compared with the 38 field boulder bars, and the geometry characteristics of the two were found to be consistent, which verified the results of this article is credible.

• *As Reviewer #1 also pointed out, the manuscript is lacking a presentation of hypotheses or a clear scientific question to answer. Currently, the closest statement to an objective is found on Lines 79-82; however, this statement sounds more like a descriptive aim rather than a clear objective addressing a gap in scientific understanding of this type of system. Please phrase your objective to answer a scientific question.*

Thank you for this suggestion. We have added a description of the scientific question and objective in the Introduction (see the 5th and 6th paragraph in *Introduction*).

• *Discussion: The discussion is somewhat approved from the previous version; however, the Discussion needs substantial improvement in order to put these results in a larger scientific context. First, the Discussion should start with referring back to the scientific objective stated in the Introduction (see previous point) in order to summarize the results for the reader and explain how the scientific objective was met (or answers hypotheses). Then you should explain how your results are supported (or not) by other studies. Currently, the Discussion only references two other studies,*

*which is not sufficient to put this study in a larger context of boulder bars or channel morphology downstream of outburst floods. These cannot be the only two studies examining morphology downstream of outburst floods. This comment relates to the first bullet point that the manuscript is lacking scientific justification and explanation of how this study connects to a wider scientific context of outburst floods, channel morphology and boulder bars, beyond this one case on the Yigong River.*

Thanks for your suggestion. We collected four field cases, which are Yigong landslide dam, Tangjiashan landslide dam, Sedongpu landslide dam, and Hongshihe landslide dam. We investigated the geometric size and distribution of 38 boulder bars in the field. And compared them with the boulder bars formed in the experiment. The results show that the experimental results are consistent. We have added the related contents in *Discussion* section.

*• L259: Minor comment about word choice: the word 'abdomen' was changed to 'waist', but this is still not the appropriate word here. Perhaps 'middle' is more appropriate?*

Thank you for this suggestion. The word "waist'" has been changed to "middle" as you suggested in the revised manuscript (See line 293 of revised manuscript for details).

*Comments of Reviewer #1:*

**General Comments**: *The manuscript by Dr. Jiang and colleagues summarizes results of an experiment investigating bar dynamics following breach of a landslide dam. The manuscript appears to be a re-working of results from a similar paper published by the same lead author in 2020 in the journal 'Landslides' (Jiang et al., 2020, cited in the manuscript). The experimental design appears sound, the experiment is well documented, and the results appear different enough from that paper to justify a separate publication. Nonetheless, the current manuscript suffers from a confusion of terminology and formative processes of the primary sedimentary body being investigated (fluvial bars), is lacking in scientific justification, and does not effectively communicate the novel scientific contribution of the experimental results. It is my judgement that the results of the experiment could make a contribution to the scientific community, but the manuscript needs very substantial revision to meet the aims and scope of Earth Surface Dynamics.*

Thanks a lot for the reviewer's comments. We agree with the reviewer's opinion that results of the experiments could make a contribution to the scientific community. In the revised manuscript, we have pointed out this manuscript's contributions to the scientific community and aims of this paper. We have added the explanations in the *Abstract, Introduction*, and *Conclusion* section. Please see the *Abstract, Introduction*, and *Conclusions* sections in the revision.

And we have re-examined the results of the experiment and confirmed that the boulder bar was formed in the experiment instead of the sandbar. We also have made

other revisions to the manuscript based on the reviewers' comments. We have marked the revised contents have been highlighted in blue in the revision.

**Specific Comments:**

*1.The use of the term 'sandbar' is ill-founded. The experiment does, in fact, use a substrate that is approximately 40 percent sand. However, because of the scale and high Froude conditions (>2), the experiment best represents a canyon, gravel bed system. The formative processes of 'sandbars' in this experimental design are entirely different than the sedimentary bodies described in lines 54 to 108 of the Introduction. In that section, there is extensive review of sedimentary bodies that are not genetically nor stratigraphically related to the sandbodies formed in these experiments which, at the scale of the experiment are gravel alternate bars. The fact that the bars in the experiment migrate in the upstream direction is evidence that the experiments are simulating Froude-supercritical (diffusive) conditions (Shaw and McElroy, 2016), whereas most of the sandbars described in the Introduction (except those formed by landslide dams) are formed by translative depositional processes. I would suggest the authors re-visit the process scaling of the experiments to re-frame and strengthen the experimental justification and basis, and the scientific contributions of the results. Kleinhans et al. (2014) and Shaw and McElroy (2016) provide excellent discussions on linkages between sedimentary processes in flumes and those in rivers.*

Thanks a lot for the reviewer's comments. The authors have discussed this comment

and agreed with the reviewer. We reviewed the experimental screen and confirmed that the bars formed in the experiment were gravel bars, which correspond to the boulder bars in the field (Wu et al., 2020; Turzewski et al., 2019). We have corrected the term "sandbar" to "boulder bar". The *Introduction* has been rewritten focused on boulder bars, the literatures have been recited, and lines 54-108 of the introduction of the original article have been revised by us to ensure accurate description of the boulder bar.

In addition, we agree the reviewer's opinion that the boulder bar in the experiment is indeed formed by the diffusion rather than translation of bedload. And we also found this phenomenon in the experiment. As shown in Figure 1, when the discharge is reduced, some gravels will stay on the river bed and hinder the advancement of the upstream flow, and part of the sediment in the flow will be deposited. As time goes by, the accumulation of sediment on the side of the boulder bar increases, and the boulder bar appears to develop upstream. We have added the explanations as the reviewer's suggestion to the 3$^{rd}$ paragraph of *Introduction* section and the last paragraph of section 3.1.

Thanks for your suggestions about experimental model design. In this manuscript, we did not design the test models based on a field prototype. However, it did not mean there is no basis for our experimental design. In fact, the experimental dam models used in the tests are satisfied with the geometry characteristics of filed landslide dam. And the composition characteristics of the materials in the test are similar to the compositions of filed landslide dam and its downstream boulder bar. In

addition, we have compared the experimental results with the field data, and have demonstrated the reliability of experimental results. Please see the *Discussion* section.

[Figure]

**Figure.1**. boulder bar obstructs the outburst flow to the river bed lower reaches

*any discussion linking the experimental results to the broader scientific literature, nor*

*does it effectively relay the importance of the results to interpretation or prediction of*

*landslide-dam outburst events.*

Thanks a lot for the reviewer's comments. We have simplified the description of the

position and size characteristics of the boulder bar in the experiment according to the

reviewer's suggestions. Please see Section 3. As your suggestion, we have added a

discussion section to compare the experimental results with four field cases. The

experimental results and field data are consistent, indicating that the experimental

results can provide references for the study of the formation and development of the

boulder bar formed by the outburst flood. And the scientific contribution was also

presented in the *Discussion* section, as follows:

**"5. Discussion**

In this paper, eight groups of landslide dam failure tests were conducted to

[revised manuscript text omitted]

**Summary:**

*The authors use a flume study to understand the effects of outburst flooding on downstream sandbar development. Different dam geometries (width and downstream slope angle) and a constant dam height were used. The upstream pool was allowed to fill and then overtop and fail under the same constant flow rate in all experiments. The authors relate bar frequency and volume to different dam geometries, and also note that bars tend to grow upstream during the experiments. The authors proceed to relate their observations to the flow hydraulics and sediment concentrations during the experiment. While the experimental set up seems reasonable, and the general result reproducible, there are several parts of the analysis that are flawed. For example, the "sandbars" are not scaled appropriately, and in fact the median grain size is gravel in the experiment. Instead, these grains are equivalent to very coarse (boulders?) grains in the field scale. The framing of the introduction and paper in general is therefore not appropriate. Further, the Froude numbers during these experiments are all supercritical, leading to spurious correlations between transport capacity and flow depth (for example, I assume that dimensionless shear stress is calculated using subcritical flow assumptions via the depth-slope product embedded in u\*). Nor is it clear how sediment concentration (figure 8) was calculated with the reference to Laursen given. Later, they use the Meyer Peter-Muller equation to calculate bedload, but, again, not considering the supercritical flow regime of the*

*experiments and the influence on energy slope as far as I can deduce. Therefore, it is difficult to interpret whether any of the results in sections 4 and 5 are valid.*

Thanks a lot for the reviewer's comments. As the reviewer concerned, the bar formed in the experiment was composed of a lot of coarser materials, which should be named "boulder bar". We agree your opinion and have corrected the term "sandbar" to "boulder bar" in the revision.

Considering the content of the original manuscript is more, and the sections 4 and 5 have caused the reviewer greater confusion, we have deleted the sections 4 and 5 in original manuscript after our careful consideration. And in the revised manuscript, geometry characteristics of the boulder bars after dam failure was added (section 4 of the revised revision), and the results of this experiment were compared with the field boulder bars in the *Discussion* section (section 5 of the revised revision), which proved the reliability of the results of this experiment. It shows that the research results of 
[revised manuscript text omitted]

**General Comment:**

*Introduction: make it clear how the background information will provide context for the results of the study. For example, the reference to Demirci et al. (2014) does not provide much insight into how these results for a coastal beach will provide context for this study. The authors could use the introduction to describe more precisely how these different previous studies relate to sandbars formed in settings 1, 2, and 3 described on lines 67-70. And then state how the sandbars in this study fit within one of those settings, or whether they are some different phenomenon related to outburst floods (as is implied). Further, the references should be more directly related to the coarse-grained alternate bars that form during the experiment, rather than sandbars.*

Thanks a lot for your comments. According to your valuable suggestion, we have corrected the term "sandbar" to "boulder bar" in the revised manuscript. Also, we have rewritten the introduction part of the article and recited references related to the boulder bar to ensure the correct citation of the references. Please see the *Introduction* section.

*Section 3: It would be very useful to have some information on grain size on the bars in this section. Much of the sediment in the experiment seems equivalent to boulders in the field case, and the coarse sediment seems to comprise much of the bar material. Even in that case, the grain size of the sediment is going to be a very important*

*factors in depositional patterns and should also be reported.*

We fully agree with the reviewer's opinion. We report the material size information of the boulder bar downstream of the river bed (section 3 in the revised manuscript). We found that the farther away from the dam, the smaller the median particle size of the boulder bar material for the boulder bar on the river bed. The experimental results are in good agreement with the Yigong flood (Turzewski et al., 2019). We have added the related descriptions and explanations in the 3$^{rd}$ paragraph of section 3. The details are as follows:

"Turzewski et al. (2019) measured the sizes of field boulder bars. They found that grain sizes of boulder bars decrease downstream. In this experiment, sediments in boulder bars after dam failure from different locations were collected. After sieving the sediments, the gradation curves of the materials were obtained as shown in Fig. 4. The figures show that the contents of fines in the compositions become much less and their mean diameters become larger than the initial sediments. It means that in the boulder bars coarse sediment tends to comprise much of the bar material. Meanwhile, the figure indicates that as the distance between the boulder bar and the dam increases, the particle diameter in the bars shows a decreasing trend. This feature is consistent with the description of Turzewski et al. (2019).

[Figure]

**Figure. 4.** Gradation curve of the boulder bar materials. Notation: U, M, D, MU, and MD, represent the boulder bar near the upstream reaches, the boulder bar near the middle reaches, the boulder bar near the downstream reaches, the boulder bar near the middle-upstream reaches, and the boulder bar near the middle-downstream reaches, respectively."

Reference:

Turzewski, M.D., Huntington, K.W., and Leveque, R.J.: The Geomorphic Impact of Outburst Floods: Integrating Observations and Numerical Simulations of the 2000 Yigong Flood, Eastern Himalaya, Journal of Geophysical Research: Earth Surface, 124, 5, https://doi.org/10.1029/2018JF004778, 2019.

*Section 4: It is confusing that sediment concentration is calculated using a reference to Laursen, with no reference to how this was done or whether we are talking about bed load or suspended load. For true sandbars, it seems that the suspended sediment component would dominate. Later in this section, the MPM formula is used for bed*

*load transport capacity, but how is u\* defined given the supercritical flow conditions.*

*I don't know if it is appropriate to use MPM without consideration of the effect of Froude number on the energy slope; this may lead to spurious negative correlations between Froude number and transport capacity.*

Thanks a lot for the reviewer's comments. it is difficult to measure the concentration and sediment carrying capacity of outburst flow exactly and timely during experimental process. So, we adopted the calculated methods to obtain the concentration and sediment carrying capacity values in the first version manuscript. However, the content of section 4 in the original manuscript may cause confusion to the readers. Therefore, we decided to delete the content of section 4 in the original manuscript. And we added the contents about geometry characteristics of the boulder bars in section 4 in the revised draft as follows:

**"4. Geometry characteristics of the boulder bars after dam failure**

In the Sec.3, we introduced formation characteristics and the geometry characteristics of the boulder bars during the dam failure processes. In this section, we will introduce the geometry characteristics of the boulder bar after the dam failure. After the dam failure, there were 25 boulder bars formed along the channel for all the tests. And it reflected the number of boulder bars was 0.4 to 1.0 times the ratio of river bed length to dam bottom length. The parameter $R$ is defined as the ratio of boulder bar length $L$ to width $W$ in Eq. (1). And the dimensionless length $L^*$ is calculated with

Eq. (2), where $L_d$ is dam bottom length.

Figure 8(a) shows the relationship between $R$ and the $L^*$ of the 25 boulder bars after the dams' failure in the experiments. The figure indicates that the values of $R$ of the boulder bars all fell within the range of 8 to 14. And, the $R$ increases with the increasing of $L^*$. However, the growth rate of $R$ decreases as $L^*$ goes by. The figures show that there is a hyperbola relationship between $R$ and $L^*$. The hyperbolic function means that $R$ would not sharply increase even become stable with the increasing of $L^*$.

$$R = \frac{L}{W} \tag{1}$$

$$L^* = \frac{L}{L_d} \tag{2}$$

Two dimensionless parameters $A_1^*$ and $A_2^*$ are defined to reflect boulder bar's area and channel cross-sectional area where the boulder bar located. They could be obtained by Eqs. (2) and (3) respectively. The relationship between $A_1^*$ and $A_2^*$ is shown in Fig. 8(b). It can be seen that $A_1^*$ increases as $A_2^*$ increases. And there is a linear relationship between $A_1^*$ and $A_2^*$. The figure suggests that the ratio of boulder bar's area to river channel cross-sectional area is approximately constant, which equals to 0.53.

$$A_1^* = \frac{A_1}{L_d^2} \tag{3}$$

$$A_2^* = \frac{A_2}{L_d^2} \tag{4}$$

[Figure]

**Figure.8.** Geometry characteristics of boulder bars after the dam failed in the experiments. (a) the relationship between length to width ratio ($R$) and dimensionless length ($L^*$); (b) the relationship between boulder bar's dimensionless area ($A_1^*$) and the cross-sectional dimensionless area of the river channel along the boulder bar ($A_2^*$)."

*Section 5: given the unknown equation to calculate sediment concentration, and uncertainty in the calculation of sediment transport capacity described above, I don't know how to interpret the results of this section.*

Thanks a lot for the reviewer's comments. After our careful discussion, we decided to delete section 5 from the original manuscript to avoid making confusion for readers.

**Line Comments:**

*29: It "is" found. . .*

Thanks for the reviewer's guidance. The line 29 in the original manuscript is related to

the Froude number. In the revised manuscript, we have deleted the content related to the Froude number. Therefore, this sentence is no longer in the revised manuscript.

*35: Exchange "reference the research on" with "can be applied to"*

Thanks a lot for the reviewer's comments. We have made corrections in accordance with the reviewer's requirements. See the line 32 of the revised manuscript for details: " Therefore, the results in this paper are credible, and can be applied to the river bed's geomorphological characteristics analysis triggered by overtopped landslide dam failure."

*41: delete "collapses" and just use "landslides"*

Thank the reviewer for this suggestion. We have made corrections in accordance with the reviewer's requirements. See line 39 of the revised manuscript for details: "Activities such as rainfalls and earthquakes often cause landslides, which block the river to form a water-retaining body similar to a reservoir dam, called a landslide dam (Takahashi, 2007; Costa and Schuster, 1988; Casagli, 2003). "

*60: "At present, much research. . ."*

Thanks a lot for the reviewer's comments. We have corrected the sentence in the

revision.

*74: throughout the introduction, I suggest replacing semicolons with periods and starting new sentences.*

Thanks very much for the reviewer's comments. We have rewritten the *Introduction* section in the revised manuscript based on your valuable suggestion.

*118-131: good concluding paragraph of the introduction*

Thanks very much for the reviewer's compliment. We used these sentences in our rewritten *Introduction*:

"Above all, there is a common academic consensus that outburst flow triggered by landslide dam failure could change the geomorphology of downstream riverbed. Although, the failure process of the dam and the hydraulic characteristics of the outburst flood, such the characteristics of breaching hydraulic graph, erosion rate and peak discharge (Morris et al., 2009; Jiang and Wei 2018; Jiang, 2019), the impact of the outburst flood triggered by landslide dam failure on the geomorphology of the downstream riverbed during the failure process and after failure is still lack of research. Boulder bar is the substance occurred during the dam failure process which is an indicator for the variation of riverbed geomorphology. What are the formation characteristics of boulder bars during the dam failure process? And what geometry

characteristics of boulder bar are during the dam failure process and after the dam failure? These questions are still not clear and should be answered. Understanding these questions is helpful for predication of riverbed landform influenced by landslide dam failure, and benefit to assessment of stream restoration and river navigation.

This paper focuses on the formation processes and the geometrical size characteristics of boulder bars in the downstream channel during and after the overtopping failure process. Firstly, through flume experiments, boulder bars' formation processes on the downstream channel under the dammed lake failure condition were reproduced. Then, based on the experimental data, the development characteristics of boulder bars' upstream and downstream edges were analyzed. Furthermore, statistical analysis of boulder bars geometrical sizes at each moment during and after the failure process, such as length, width, height, volume and area of boulder bar, had been carried out to obtain boulder bars' size characteristics. Finally, compare the distribution and geometry characteristics of the boulder bar formed in the experiment and field to verify experiment results' reliability. The results can be applied to the river bed's geomorphological characteristics research affected by the outburst flood triggered by landslide dam failure. And also, this paper provides a large number of experimental and field boulder bars' data reference to the analysis of the erosion and accumulation characteristics of the downstream river channel."

*156: spelling: "gravels"*

Thanks a lot for the reviewer's comment. We have revised the spelling of the "gravels":

" The dam materials used in this study were mixtures of sand and gravels. "

*156-157: Was there any dimensional scaling of the grain size? What would this sediment size correspond to in a field setting?*

Thanks a lot for your comments. Although, there was not any dimensional scaling of the grain size, the type of materials used in the tests was similar to the filed, and the materials could ensure the overtopped failure mode happen for the tests. Most of the landslide dams in the field are mixtures of fine particles and coarse particles, therefore, we selected mixtures of sand and gravels as the experimental materials. With reference to Vallejo and Mawby (2000) and Wan and Fell (2008), considering the limitations of the experimental flume space and the size of the dam model, the experimental material adopted a median particle size of 3.8mm mixtures of sand and gravels to improve the possibility of overtopping failure of dam model. Therefore, the selection of dam model and material in this experiment could meet the experimental requirements.

*180: Were the balls buoyant in the flow?*

We are honored to be able to answer the reviewer's confusion. The balls used in the experiment have a small mass and can float on the flow surface, and can be used to measure the flow velocity.

*188: Are you only able to measure the height along the flume wall? Rather than the average height across the channel?*

We are honored to answer the reviewer's questions. We could measure the height along the flume wall timely and conveniently during the dam failure process. Although, the height of other positions may be different with the section along the flume wall, the difference is small based on the observed data. So we selected the boulder bar's section along the flume wall as concerned positions of the boulder bar and measured the maximum height of the boulder bar of these positions.

*220: Are these sandbars or gravel bars? They look to be dominated by the coarse*

*fraction in the photos.*

Thanks a lot for the reviewer's reminder that gravel bars were formed in the river bed during the dam failure in the experiment, which corresponds to the boulder bars formed by the outburst flood in the field. We have also corrected this concept in the revised manuscript.

*227-229: This sounds like alternate bar formation, for which there is significant literature that was not discussed in the introduction.*

Thanks a lot for the reviewer's comment. As the reviewer mentioned, we have changed the concepts of "sandbar" to "boulder bar". In the revised manuscript we have rewritten the *Introduction* on "boulder bar".

*314-316: I don't understand why a smaller discharge would lead to a larger bar spacing. Please elaborate.*

It is a great honor for us to explain this phenomenon to the reviewer. Maybe we did not describe it clearly that the "discharge" is "inflow discharge" upstream the dam. We have clarified this term in the revision. Although the inflow discharge in the experiment is small, but the stored water volume behind the dam may be large. When the water volume in front of the dam is large enough, the landslide dam will be

overtopped, and the dam will be failure very quickly. Water will be released in a short time, and the outburst discharge may be large (Jiang and Wei, 2020; Carrivick 2010; Jiang and Wei, 2018). We have deleted the confusing sentences.

After careful consideration, we decided to delete the calculation of the concentration in the revised manuscript. There are no contents related to concentration in the revised manuscript.

*422: Spelling: abdomens? I think a different word was intended.*

Thanks a lot for the reviewer's suggestion. We followed the editor's suggestion to replace "abdomen" with "middle" in the revised manuscript (see line 293 in the revised manuscript).

*444: Were the concentration calculations using Laursen based on bedload as well?*

Thank you very much for the comment. After careful consideration, we decided to delete the calculation of the concentration in the revised manuscript. There are no contents related to concentration in the revised manuscript.

*451: equation 4 is not correct – need to square u\* in the numerator*

Thanks a lot for the reviewer's correction. We have deleted equation 4 in the revised manuscript.

*473-477: Not sure I agree with this logic. These Froude numbers are well over 1 in the supercritical regime. The shear stress as calculated is lower at higher Froude numbers because it will be shallower, but the velocity will actually be even greater.*

Thanks a lot for the reviewer's comment. This comment is very helpful to us. We all agree with the reviewer's point of view. After the authors' discussion, we have deleted this part in the revised manuscript.

*Figure 3: It looks like these are essentially alternate bars forming in a straight flume channel – you state that this is similar to the field setting, but are the bar locations sometimes also controlled by the presence of obstructions?*

Thanks a lot for the reviewer's comment. The question raised by the reviewer is very valuable. Boulder bars' locations are sometimes controlled by the presence of obstructions. The river bed downstream terrain conditions of the field landslide dam are more complicated. In the lab, we simplified the experimental conditions. For example, we simplified the channel shape and omitted the obstacle in the channel. The straight channel used in the tests is a common simplified model (Jiang and Wei, 2020; Chen et al., 2015). It is convenient for us to use the straight channel model to summarize the characteristics of boulder bar's formation and development. When the river bed in the experiment is not of equal width and straight, it is not conducive to drawing a general rule. In addition, the experimental dam model designed according to the method of Zhou et al., (2019), which can reflect the actual characteristics of the field landslide dam. And according to the research of Vallejo and Mawby (2000) and Wan and Fell (2008), the materials for this experiment can be meet the experimental requirements.

*Figure 4: I wonder if this figure could be simplified to focus on the key points in the discussion of the figure that describe the 3 variations of response*

Thanks a lot for the reviewer's comment. Figure 4 in the manuscript shows the change of the position of the boulder bar during the dam failure process. Figure 4 can very intuitively and vividly express the change of the position of the boulder bar over time. Therefore, if possible, we would like to retain the contents of Figure 4. But the format of the figure has been changed as follows:

[Figure]

(a)

[Figure]

(b)

[Figure]

(c)

[Figure]

(d)

[Figure]

(e)

[Figure]

(f)

[Figure]

(g)

[Figure]

(h)

**Figure. 5.** The boulder bars' locations during the dam failure process. Notation: (a) to (h) represent the boulder bars' locations for T1-T8 tests, respectively. The red lines in the figure represent the boulder bars, and the orange rectangles represent the channels. The numbers at both ends of the red lines represent the distances between the upstream and downstream edges of boulder bars and the dam toe.

*Figure 5: This is a complex of a figure relative to its discussion in the text; the scale bar doesn't allow us to see much of a trend except for length. There is not consistency in the labeling scheme (dots for length, triangles for width, for example; same colors for the same model runs).*

Thanks a lot for the reviewer's comment. According to the reviewer's suggestion, we have modified Figure 5 in the manuscript. We hope that the revised figure can satisfy the reviewer and the readers. The revised figure is following:

[Figure]

**Figure. 6.** The lengths, widths, and heights of the boulder bars: (a) sizes of the boulder bars near the upstream reaches; (b) sizes of the boulder bars near the middle reaches; (c) sizes of the boulder bars near the downstream reaches. Notation: L, W, and H represent the length, width, and height of the boulder bar, respectively. i represents the Ti experiment. For example, MUL6 indicates the length of the boulder bar near the middle-upstream reaches for the T1 test.

*Figure 6: Same comment as figure 5 with regard*

Thanks a lot for the reviewer's comment. This suggestion is very helpful to us. We have modified Figure 6 in the manuscript as suggested by the reviewer, as shown below:

[Figure]

**Figure. 7.** Volumes of boulder bars. Notation: UVi, MVi, DVi, MUVi, MDVi represent the volume

of the boulder bar near the upstream reaches,the boulder bar near the middle reaches, the boulder

bar near the downstream reaches, the boulder bar near the middle-upstream reaches, and the

boulder bar near the middle-downstream reaches, respectively. For example, UV1 means the

boulder bar's volume near the dam toe of the T1 test

---

## Author Response (AR3)

**Responses to editor and reviewers**

Dear editor,

Thanks very much for taking your time to review this manuscript entitled "The formation and geometry characteristics of boulder bars due to outburst flood triggered by the overtopped landslide dam failure" (Esurf-2020-92). I appreciate all your and reviewers' comments and suggestions! Those comments are constructive for us to revise and improve our paper. We have read these comments carefully and tried our best to revise and improve the manuscript. The revised contents have been highlighted in blue in the revision. The followings are the point-to-point response for each comment. We hope our revision will make the manuscript acceptable for publication. Thanks again!

Best regards,

Xiangang Jiang

*suggestions of Editor:*

*I have now reviewed two reviews of your revised manuscript, one of whom reviewed the original manuscript. Although Referee #3 has some serious concerns regarding congruency of the experimental design with the study objectives, I believe that you should be able to address this concern through more careful wording of the objectives and discussion of the results. In addition to addressing this major concern, please address the additional minor comments from both reviewers before the manuscript is ready for publication.*

Thanks for your suggestion. We have added the relative explanations in section 2.1 and

Discussion in the revision. Please see 146 to 156 lines of page 8 and 358 to 360 lines

of page of 21.

**Response to the Reviewer 2**

*Line 17: check that you have written "field" rather than "filed" where appropriate in*

*the text.*

Thanks a lot for the reviewer's comment. We have changed "filed" to "field" in the

revision. Please see the line 17 of page 1.

*Figure 6: consider a log scale for the y-axis so that the width and height data are easier*

*to see.*

Thanks a lot for the reviewer's suggestion. We have adjusted Figure 6 as follows:

[Figure]

**Figure. 6.** The lengths, widths, and heights of the boulder bars: (a) sizes of the boulder bars near the upstream reaches; (b) sizes of the boulder bars near the middle reaches; (c) sizes of the boulder bars near the downstream reaches. Notation: L, W, and H represent the length, width, and height of the boulder bar, respectively. i represents the Ti experiment. For example, MUL6 indicates the length of the boulder bar near the middle-upstream reaches for the T1 test.

*Figures 8 and 10 are a nice addition to this study: consider plotting the flume data on the field data plot in figure 10 to show how well they compare*

Thanks a lot for the reviewer's suggestion. We have plotted the flume data on the field data plot in figure 10. It shows the two groups of data are compared well.

[Figure]

**Figure.10.** Geometry characteristics of boulder bars after the dam failed in the field. The experimental data are also plot in the figure to compare to the field data. (a) The relationship between boulder bar length to width ratio ($R$) and dimensionless length ($L^*$); (b) The relationship between boulder bar's dimensionless area ($A_1^*$) and the cross-sectional dimensionless area of the river channel along the boulder bar ($A_2^*$).

**Response to the Reviewer 3**

*1.The materials of riverbed is quite different from that of landslide dam, thus an inappropriate design is that the riverbed consisted of the same material as the landslide dam;*

Thanks a lot for the reviewer's comments. There is indeed difference between the materials of the field riverbed and dam. However, in the existing researches, the difference of materials between the two has not been quantitatively analyzed so far, and there is no clear description on this subject. Moreover, in different areas, the materials of landslide dam and riverbed may be significantly different. Therefore, in the present

experiments, it is considered that the dam material and the river bed material are the same.

In lines 146-148 of the revised manuscript, we have added explanations as: "While the materials of riverbed are different from that of landslide dam, it is hard to find a general description of the difference. Thus, we designed the materials of riverbed and landslide dam the same for present experiments."

And, we have also added the related contents in the Discussion section in the revision.

*2. The initial location (or distribution) of coarse particle within landslide dam is unclear, and this factor can have an important influence on dam failure and boulder bars formation;*

Thank you very much for the comments. As the reviewer's comment, the initial position (or distribution) of the coarse particle within the landslide dam has an important influence on dam failure and boulder bars formation. We agree with the reviewer's opinion. However, there is still no quantitative and accurate description about the distribution of the coarse particles within the dam. To simplify the experimental conditions, homogenous dams and riverbeds were set in the experiments. We have added the contents in lines 148-152 of the revised manuscript as follows "Moreover, the compositions of field dam and riverbed can be heterogeneous, i.e. the distribution of coarse particle within landslide dam is inhomogeneity, there is still no quantitative representation of the heterogeneity. Therefore, the coarse particles and fines were mixed

uniform, which means the distribution of coarse particles were homogeneous."

Also, the related contents have been added in the Discussion section.

*3. Channel morphology that probably has a clear impact on the formation process and geometry characteristics of boulder bars is not considered. The results based on the experiment with only a river type are unconvincing, e.g. two dimensionless parameters;*

Thank you for your comment. The authors agree the reviewer's opinion. However, the topography of river channels in the field is complex and diverse. To reveal the fundamental characteristics of boulder bars' formation process and geometrical characteristics, the experiments did not consider the real channel morphology. Instead, a straight and flat river channel is adopted. This "idealized" model is conducive to reveal the boulder bar formation process and geometrical characteristics. We have added an explanation in the revised manuscript, please see lines 152-156.

*4. The flow direction in fig. 5 should be marked.*

Thanks a lot for the reviewer's comments. We have added the flow direction mark in fig. 5 according to your suggestion in the revision.